# Evidence for and significance of the Late Cretaceous Asteroussia event in the Gondwanan Ios basement terranes

Sonia Yeung[1], Marnie Forster[1], Emmanuel Skourtsos[2], Gordon Lister[3]

[1] Structure Tectonics Team, Research School of Earth Sciences, Australian National University, Canberra, 2601 Australia

[2] Section of Dynamic, Tectonic and Applied Geology, Department of Geology and Geoenvironment, National and Kapodistrian University of Athens, Athens 157 72, Greece

[3] W.H. Bryan Mining and Geology Research Centre, Sustainable Minerals Institute, University of Queensland, Australia

*Correspondence to*: Sonia Yeung (HoSonia.Yeung@anu.edu.au)

**Abstract.** The Late Cretaceous Asteroussia event as recorded in the Cyclades is a potential key to the tectonic evolution of Western Tethys. Microstructural analysis and $^{40}Ar/^{39}Ar$ geochronology on garnet-mica schists and the underlying granitoid basement terrane on the island of Ios demonstrates evidence of a Late Cretaceous high pressure, medium temperature (HP–MT) metamorphic event. This suggests that the Asteroussia crystalline nappe on Crete extended northward to include these

Gondwanan tectonic slices. In this case, the northern part of the Asteroussia nappe (on Ios) is overlain by the terrane stack defined by the individual slices of the Cycladic Eclogite-Blueschist Unit, whereas in the south (in Crete) the Asteroussia slices are near the top of a nappe stack defined by the individual tectonic units of the external Hellenides. This geometry implies that accretion of the Ios basement terrane involved a significant leap of the subduction megathrust (250-300 km) southward. Accretion needs to have commenced at or about ~38 Ma, when the already partially exhumed slices of the Cycladic Eclogite-

Blueschist Unit began to thrust over the Ios basement. By ~35-34 Ma, the subduction jump had been accomplished, and renewed rollback began the extreme extension that led to the exhumation of the Ios metamorphic core complex.

## 1 Introduction

A terrane stack accreted on the northern edge of the Tethys Ocean during the episodic closure of this ocean basin. Several of these tectonic slices now outcrop on the island of Ios, in the Cyclades, Greece (e.g., Durr et al., 1978; Andriessen et al., 1986;

Forster & Lister, 1999a, b; Ring et al., 2007; Forster & Lister, 2009). Tectonic slices in the Cycladic Blueschist Unit were subject to high pressure metamorphism, and later juxtaposed against tectonic slices of Hercynian continental basement. How this juxtaposition occurred remains controversial (Ring et al., 2007; Huet et al., 2009; Forster and Lister, 2009, and references

therein). One of the competing hypotheses is that a succession of tectonic mode switches took place, with episodes of crustal shortening prior to each of a succession of accretion events. Each accretion event appears to have been followed by an episode of crustal extension (Forster and Lister 2009). Extension following the accretion events that occurred later in this history caused core complex formation, including the formation of the first-recognised Aegean metamorphic core complexes (Lister et al. 1984). The hypothesis that is the main contender as an opposing point of view is that this did not occur, and the Cycladic Blueschist Unit continually extruded during the long history of Alpine convergence (in the so-called orogenic phase, Huet et al., 2009). However, Forster and Lister (2009) unequivocally demonstrate that a discrete succession of exhumation events juxtaposed the Hercynian granitoid basement (identified as the Ios basement terrane in this paper) against the overlying Cycladic eclogite-blueschist slices, so this hypothesis (at least in its present form) is not tenable. Either, the Cycladic Blueschist Unit must have been earlier over-thrust (at ~38 Ma, Forster and Lister, 2009), and largely eroded: or, alternatively, at about ~38 Ma, the Ios basement terranes must have begun to subduct beneath an already largely-exhumed Cycladic Blueschist Unit. In this case, subduction must have continued until these Gondwanan terranes were accreted at ~35 Ma. Their subsequent crustal extension involved a succession of extensional ductile shear zones and later-formed detachment faults.

Dispute arises in part because insufficient information is available as to the details of the timing and thermal evolution of individual rock units, in particular those in the Ios basement terranes that are the focus of this paper. In the extrusion model the Ios basement terranes are over-ridden as the result of thrust-induced extrusion, with deep crustal materials extruded above thrust faults that operated under continuous plate convergence, with little to no horizontal stretching (Ring et al., 2007; Huet et al., 2009). In the tectonic mode switching model, a tectonic shuffle zone must have been created in the upper levels of the Ios basement terranes, and this must have been later truncated during detachment faulting. Multiple shuffling events are implied by the several switches between horizontal shortening and horizontal stretching triggered by roll back (Lister et al., 2001; Forster and Lister, 2009; Forster et al., 2020). The marked contrast in the detail required by these competing hypotheses makes it evident that a significant knowledge gap exists in understanding the succession of discrete deformation and metamorphism events in the upper structural levels of the Ios basement terranes, in particular those that occurred prior to Alpine deformation

and separately those that occurred prior to exhumation of the Ios basement terrane (Forster and Lister, 2009; Lister and Forster 2016; Yeung, 2019; Forster et al., 2020). This research project was undertaken, in order to begin to remedy this deficiency.

Early workers assumed that, prior to Alpine time, the Ios basement was affected only by Hercynian deformation and metamorphism, based on age data from hornblende and zircon (Andriessen et al., 1987). However, white mica deformation fabrics in the Ios augengneiss core consistently yield $^{40}Ar/^{39}Ar$ ages of ~70-80 Ma (Forster and Lister, 2009). This led us to investigate the possibility that the Ios basement terrane that was made up of garnet-mica schist and augengneiss could be part of the Asteroussia nappe (c.f. Be'eri-Shlevin et al., 2009). Previous interpretation of such age data (e.g., Andriessen et al.,

1987, Baldwin and Lister, 1998) considered only the effects of 'excess argon' or 'mixing' and suggested that the apparent Late Cretaceous ages were the result of the Hercynian (~ 300 Ma) argon population mixing with Cenozoic (~ 50 Ma or younger) gas population. However, if this was the case, precisely defined Frequently Measured Ages (FMAs) would not exist in age probability plots. Therefore we were led to consider that the 70–80 Ma date reported in the structurally deepest augengneiss of the Ios lower plate was in fact the characteristic age of the 'Asteroussia event'.


To progress, we need to demonstrate that the effects of such an event can be distinguished in the complex history of deformation and metamorphism (and fluid alteration) experienced by these rocks. Therefore, we re-examined outcrops in the north-west corner of the basement terranes on Ios, in an attempt to determine the significance of the previously reported 70-80 Ma ages. We combined a field study with microstructural analysis and $^{40}Ar/^{39}Ar$ geochronology to address: i) the character and location

of micro-deformation structures with late Cretaceous age; and ii) the time relations between various metamorphic and deformation events. Our study identified relics of earlier fabrics in low strain zones that 'survived' later shear zone operation. $^{40}Ar/^{39}Ar$ geochronology on these fabrics demonstrate Late Cretaceous high-pressure metamorphism, specifically with the growth of phengitic mica in the augengneiss terrane and the overlying garnet-mica schist. The high retentivity of argon in phengitic white mica (Forster and Lister, 2014) allowed these ages to survive the thermal effects of the later Alpine history.

## 2 The Asteroussia Nappe

Late Cretaceous-aged metamorphic events were first reported from small klippen outcropped near the Asteroussia mountains in Crete, and later in various Cycladic islands (e.g., Be'eri-Shlevin et al., 2009; Dürr et al., 1978; Seidel et al., 1976). This unit is identified as the Asteroussia nappe, positioned near or at the top of the Aegean terrane stack (except where klippen of Cycladic blueschist occurred above the unit), and reflects the imprint of metamorphism in the time range 70–80 Ma (Bonneau, 1972; Bonneau, 1984). The newly defined terrane was to be characterised as having Cretaceous high-temperature low-pressure (HT-LP) metamorphic assemblages associated with granitoid intrusions with peak metamorphic conditions in the Late Cretaceous, at ~70 Ma (Dürr et al., 1978; Langosch et al., 2000; Patzak et al., 1994; Seidel et al., 1976). Table 2 shows data from other researchers who then reported Late Cretaceous ages in outcrops occurring as small klippen on various Cycladic islands (Tinos, Andros, Syros, Donoussa, Ikaria, Nikouria and Anafi) as well as in further outcrops on Crete (Avigad and Garfunkel, 1989; Be'eri-Shlevin et al., 2009; Bröcker and Franz, 2006; Dürr et al., 1978; Langosch et al., 2000; Patzak et al., 1994; Seidel et al., 1976). In turn this led Be'eri-Shlevin et al. (2009) to note that although published Rb-Sr dates on amphiboles from the Asteroussia nappe range from ~45–85 Ma, the dates cluster at ~70 Ma. These authors therefore extended the areal extent of the Asteroussia nappe to cover a north-south distance of ~300 km (Fig. 1).

The geometry of the Asteroussia nappe is complex, however, with notable local variations when comparing examples from Crete with examples in the northern Aegean Sea. For instance, on Crete a Late Cretaceous event was reported in metapelites which correlates in age with the complex upper unit of the terrane stack in some areas of the Cyclades (Avigad and Garfunkel, 1989; Be'eri-Shlevin et al., 2009; Bröcker and Franz, 1998; Bröcker and Franz, 2006; Pe-Piper and Photiades, 2006). However, the Asteroussia outcrops in Crete are small tectonic klippen (up to 10-15 km wide) with poor lithological and structural correlations (Dürr et al., 1978; Seidel et al., 1976; Seidel et al., 1981). In contrast, in the Cyclades, island-scale structural models involve two to four tectonic slices, and there are reasonable correlations that can be made across the entire archipelago. Nevertheless, some examples in the Cyclades involve meta-ophiolites and mélange zones that first underwent blueschist facies, and then were overprinted by retrograde greenschist facies metamorphism. In other cases, Asteroussia klippen overlie high-pressure metamorphic rocks from the Cycladic Blueschist Unit (e.g., Avigad and Garfunkel, 1989; Be'eri-Shlevin et al., 2009; Bröcker and Franz, 1998; Bröcker and Franz, 2006; Pe-Piper and Photiades, 2006; Ring et al., 2003). Ios stands out in that the Asteroussia ages have been obtained from the lowermost structural slices in the terrane stack, with clear proof that these are Gondwanan in their affinity (Keay and Lister, 1998). Hence, although we agree with Be'eri-Shlevin et al. (2009) that the different Cycladic outcrops in the north (Andros, Tinos, Syros, Ikaria) and in the south (Ikaria, Donoussa, Nikouria, Anafi) are

part of an extensive Asteroussia nappe that once extended northward from Crete, this observation requires the Asteroussia nappe in its entirety to have been Gondwanan in its origin. Tectonic shuffling involving large horizontal relative motions is well capable of explaining the observed complexity, but only if the terrane was first accreted in its entirety while subject to out-of-sequence thrusting and then affected by extreme crustal extension.

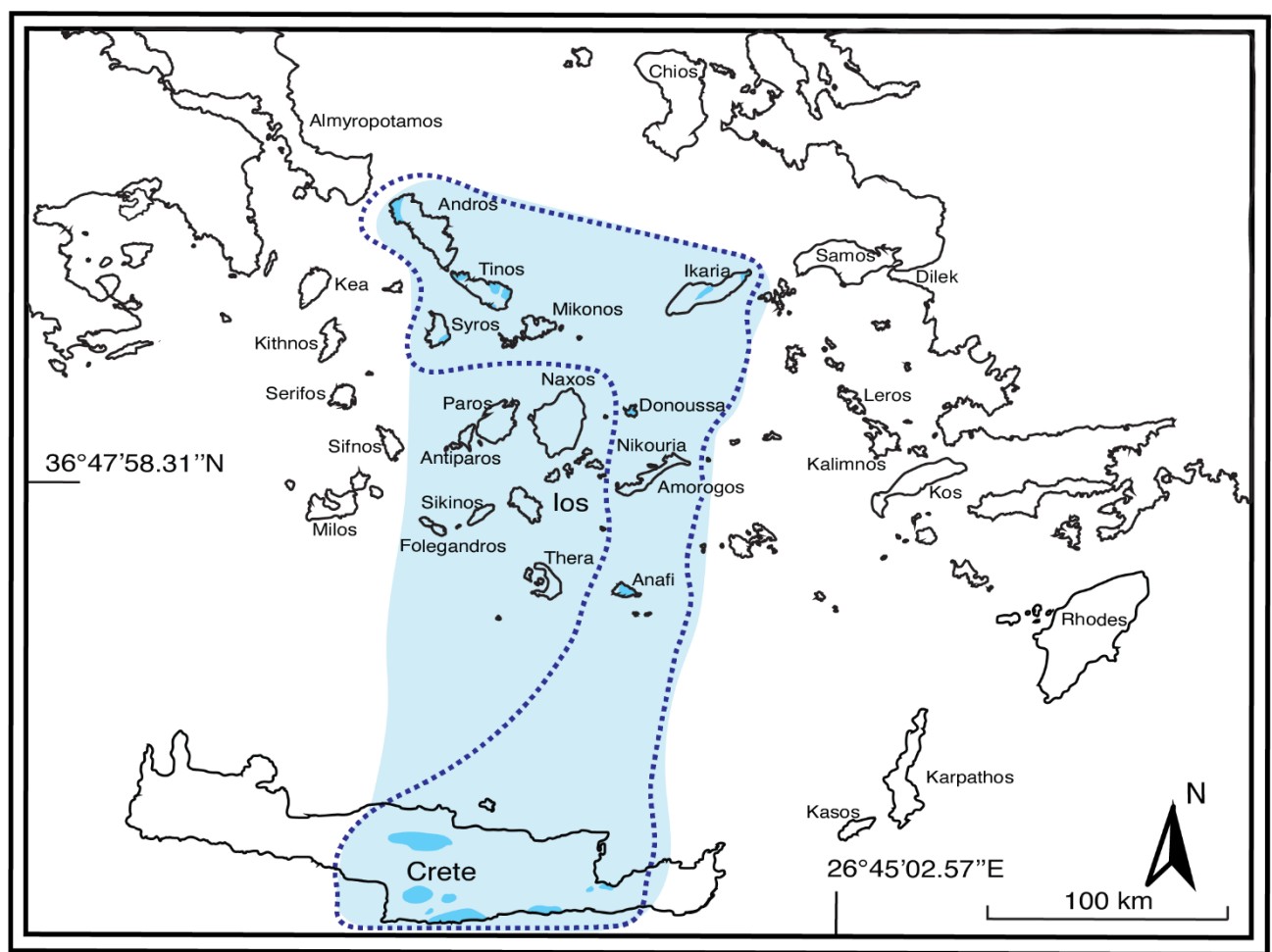

**Figure 1: Map of the Cyclades and Crete, dotted line illustrates published extent of the Asteroussia nappe: area enclosed with the dashed line includes outcrop localities with late Cretaceous age (with information retrieved from Be'eri-Shlevin et al., 2009). The area shaded in light blue is the revised areal extent of the Asteroussia nappe suggested in this paper.**

| Van Der Maar, 1979 | Vandenberg and Lister, 1994 | Forster and Lister, 2009 | Correlation of TSDs developed in this study |
|---|---|---|---|
| Pre-Hercynian granite bodies intrustion | $S_0$ layering | TSDs for earlier events not recorded | granitoid intrusion |
| $M_0$ Pre-Alpine Hercynian amphibolite facies ~295-305 Ma | $D_1$ deformation event: relicts of deformation preserved as $S_1$ microlithons in $S_2$ | | $S_3//S_{LL}$ $\Delta$grt (large porphyroblast) $S_9//S_{LL}$ $\Delta$bt (wrapping grt) |
| | | | Sdiff (Scren) |
| | | | Vein qtz ($//S_{LL}$) Alteration $\Delta$rt $\Delta$wm $\Delta$py $\Delta$hem |
| $M_1$ Eocene HP–MT eclogite-blueschist facies ~ 42-59 Ma | $D_2$ folding event (crustal shortening) formation of $F_2$ folds with: - $S_2$ axial plane cleavages - N-S oriented $L_2$ lineation | $\Delta_{1B}$ growth event in eclogite-blueschist unit (EBU), 52-53 Ma, omphacite + jadeite | porphyroblast growth event $\Delta$grt (large porphyroclast) |
| | | post $\Delta_{1B}$ SZ operation, 49-53 Ma | |
| | | $F_R$ (recumbent folding) | $F_R$ Sps $F_U$ (refolding) |
| | | $\Delta_{1C}$ porphyroblastic event in EBU, 43-45 Ma, epidote+glaucophane+garnet+mica | porphyroblast growth event $\Delta$grt (invading foam textures in quartz aggregates) |
| | | post $\Delta_{1C}$ SZ operation, 40-44 Ma | Dsz $\Delta$wm (recrystallised fabric) |
| | $D_3$ recumbent folding event (crustal shortening) formation of $F_3$ folds which: - folded $S_2$ and $L_2$, - develops $S_3$ differentiation crenulation in axial zone | $F_R$ | $F_U$ Sps |
| $M_2$ Oligo–Miocene greenschist facies ~ 25-16 Ma | | $\Delta_{1D}$ porphyroblastic event in EBU, 34-35 Ma, transitional blueschist-greenschist facies biotite+garnet+mica | $\Delta$grt light core of smaller, zoned porphyroblasts $\Delta$grt dark rim of smaller, zoned porphyroblasts |
| | $D_4$ SCSZ operation (crustal extension) forms N-S oriented $S_4$ lineations | post $\Delta_{1D}$ SCSZ operation, 35-30 Ma | ^ ^ DSCSZ (south-directed) ^ R (rotation of porphyroclasts) $\Delta$wm (recrystallised fabric) (S-C' plane fabrics) SZ (top-S) $\Delta$qtz (mantle) ^ |
| | | | Freclined |
| $M_3$ Late Miocene Granitoid intrusion (LP-HT contact metamorphism) ~ 22-10 Ma | | | $F_I$ $F_U$ (defined by qtz vein) (fold limbs recognised as boudinage) Sps |
| | | intense North-sense SZ operation, 25-29 Ma, rapid extension ~ 25 Ma with $\Delta_{2A}$ and $\Delta_{2B}$ mineral growth event under greenschist facies | ^ ^ DNCSZ (north-directed) N $\Delta$qtz (porphyroblast) $\Delta$wm (recrystallisation) SZ (top-N) R $\Delta$qtz (mantle) ^ |
| | $D_5$ Late stage warping formation of $F_5$ folds with $S_5$ axial planar fabrics | fluid activities hydrothermal sericite growth over K-spar | Pegmatite intrusion Dnormal faulting |

**Figure 2: Correlations between the results of previous work and the Tectonic Sequence Diagrams (TSDs) developed in this paper. The results imply that parts of the garnet-mica schist have been shared some parts of history of high-pressure/low-temperature metamorphism as occurred in the lowermost schist units of the Cycladic Blueschist Unit.**

## 2.2 The Asteroussia event on Ios

The terrane stack outcropping on Ios involves thin tectonic slices separated by island-scale deformation structures such as detachment faults and/ or ductile shear zones (e.g., Forster and Lister, 2009; Forster et al., 2020). All terranes experienced Early Oligocene stretching (Fig. 2) and were variably affected by the south-directed South Cyclades Shear Zone (SCSZ, $D_2$ in other publications) (e.g., Forster and Lister 1999a, b; Forster and Lister, 2009; Forster et al., 2020; Huet et al., 2009; Huet et al., 2011; Ring et al., 2007). An Eocene-Oligocene high-pressure terrane (the Cycladic blueschist unit, upper plate of the Ios Detachment) overlies the Gondwanan Ios basement terrane. There are three tectonic slices in the basement, each recording slightly different metamorphic histories (refer to maps in supplementary materials). The top-most Port Beach tectonic slice contains two lithologies (garnet-mica schist above augengneiss) and lies immediately beneath the Ios detachment (Forster and Lister 1999a). Beneath this tectonic slice is the ~500-meter-thick garnet-mica schist unit (of variable thickness across the island) at mid-structural level and the structurally deepest augengneiss core of the Ios metamorphic core complex (e.g., Andriessen et al., 1987; Baldwin and Lister, 1998; Forster and Lister, 2009; Vandenberg and Lister, 1996).

Figure 1 shows the areal extent of the Asteroussia terrane based on Be'eri-Shlevin et al. (2009), with the shaded area indicating the revised extent according to what we report in this paper. Importantly, this study recognises the exhumed Asteroussia terrane north of Crete, in the Ios basement terrane across the terrane stack, which enabled the identification of a subduction jump that is impossible without a tectonic mode switch. If correct, this is a significant modification implying a widespread metamorphic event in northern Tethys during Late Cretaceous time, including most of the Cycladic islands as reported in this paper and across the Mediterranean region (Altherr et al., 1994; Be'eri-Shlevin et al., 2009; Langosch et al., 2000). Moreover, if the extent of the basement terrane is as large as indicated in Figure 1, its accretion to the modern terrane-stack in latest Eocene time implies a southward jump of the subduction zone megathrust exceeding 250–300 km (Figs. 3 and 4). Potentially these terranes were autochthonous, with their final accretion involving a period of flat slab subduction followed by the initiation of a new subduction zone (Figs. 4c-d). Subsequent rollback can then stretch the Cycladic crust, explaining the variation in crustal thickness from ~32 km thick beneath the Cyclades, to 18 km beneath the Sea of Crete, and ~ 30 km beneath Crete (based on Makris and Vees, 1976 and Makris et al., 2001). Moreover, the subduction jump is able to explain the formation of core complexes during extreme extension caused by rollback after accretion, and the later Oligocene-Miocene magmatic event that is observed across the Cyclades.

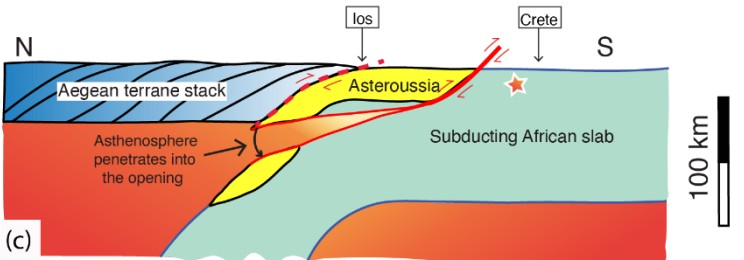

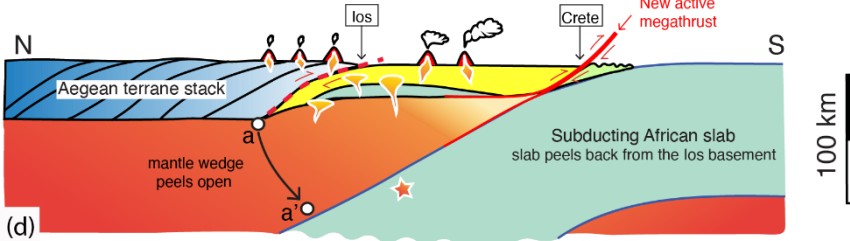

**Figure 3: Schematic cross-sections illustrating the slab-peel model. The orange star illustrates the relative movement of subducting material across time: (a) the Gondwanan Asteroussia terrane arrives at the subduction zone; (b) as it accretes to the terrane stack, the Asteroussia terrane underplates the youngest slice of the terrane stack; (c) the subduction megathrust jumps ~300 km southward, slicing the Asteroussia terrane from the subducting African slab, while subduction goes on and the slab continues to peel away from the 'buoyant' Asteroussia terrane. Asthenosphere penetrates to the dilating megathrust (d) while the subducting African slab**

**continues to roll back and the break widens. Melting of the uplifting asthenosphere causes extensive magmatism.**

**Figure 4: The slab necking-drop off model proposed in this paper: (a) the Asteroussia terrane arrives at the Aegean terrane stack; b) the subduction zone jams, so the megathrust leaps southward while the African slab begins to neck and roll back; (c) by ~ 32 Ma the subduction jump is accomplished, breaking the stretching slab; (d) a new subduction zone develops in the south, with fluids rising from the slab causing melting. Rollback of the new formed slab triggers extreme extension across the Aegean, exhuming metamorphic rocks and forming the Ios metamorphic core complex.**

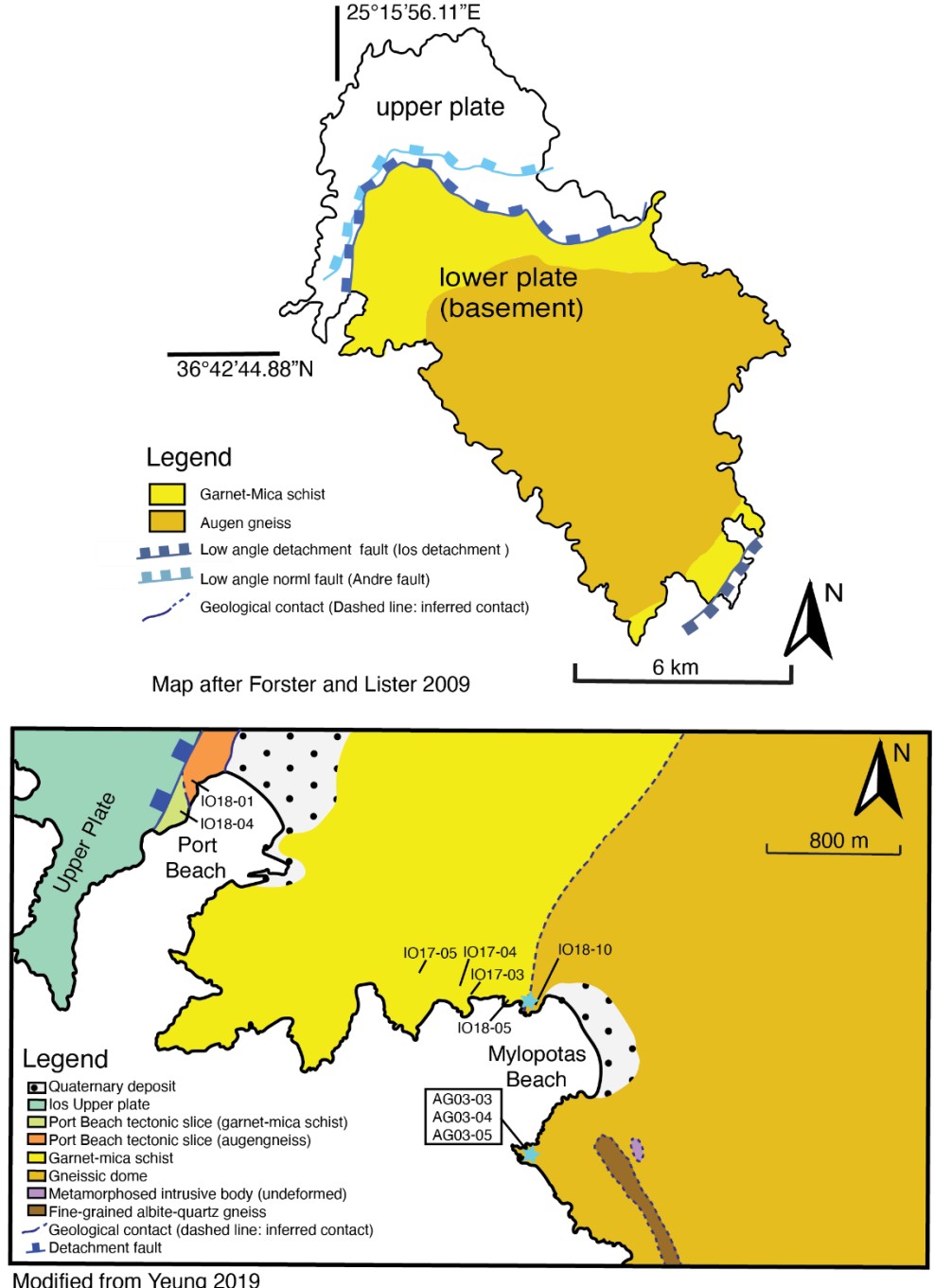

**Figure 5: Top: island-scale map illustrating major lithologies. Bottom: Detailed map of study area with sample collection sites, the entire area is affected by the broad, south-directed South Cyclades Shear Zone (SCSZ). Locations observed with overprinting narrow, north-directed shear zones are indicated by blue stars, diagrams after Yeung (2019). Detailed structural maps in supplementary materials.**

## 3 Microstructural and mineral chemistry analyses across three tectonic slices in the Ios basement terrane

Seven samples were selected (Table 1). The augengneiss (IO18-01) was collected from the structurally shallowest Port Beach tectonic slice, which has a pervasive south-directed shear fabric, with many intensive S-directed shear bands. Four samples were collected (IO17-03, IO17-05, IO17-04, IO18-05) from the deformed garnet-mica schist that underplates the Port Beach tectonic slice, each collected from different depth of this structurally mid-level tectonic slice, which preserves different stages of the South Cyclades Shear Zone (SCSZ) operation. Two samples (AG03-03, AG03-05) were examined from the upper levels of the structurally lowest unit, the augengneiss core. Results from these samples were first reported by Forster and Lister (2009) but were re-examined to establish the association between microstructure and their Late Cretaceous ~ 70-80 Ma ages.

Previous structural studies in the Ios basement generally did not recognize the presence of high-pressure rocks in the Ios basement and its tectonic history is generally identified as the $M_0$ event (Vandenberg and Lister, 1996; Baldwin and Lister, 1998; Forster and Lister, 1999b, etc.). However, the history of deformation and metamorphism in these rocks is more complex than such simple notations imply. Therefore we applied the method of tectonic sequence diagrams (TSDs) presented in Forster and Lister (2008) and Forster et al. (2020) to document the effects of the succession of pre-Alpine to Oligocene metamorphic episodes that can be observed (Fig. 2). The sequence of metamorphic mineral growth and deformation events is consistent from place to place throughout the entire shear zone carapace of the exhumed Ios basement, with the order of mineral growth episodes tied to different fabrics produced during ductile shear zone operation and/or pure shear ductile stretching of the rock mass. Figure 2 compares the results of this analysis with the traditional $D_1$, $D_2$,…$D_n$ method. The detail of relative time constraints could be accurately delineated using these TSDs.

TSDs tie metamorphic evolution to the sequence of fabric-forming events and to the processes that took place during the microstructural evolution, and are therefore critical in enabling the link between the results of $^{40}Ar/^{39}Ar$ geochronology to the detail of microstructural observations. We were able to link dates to specific deformation fabric and mineral growth events, and thus demonstrate that some of these relict fabrics preserved remnant microstructures from earlier pre-Alpine deformation

events. The effects of a high-pressure late Cretaceous event are evident in these relict fabrics in the Ios basement, so we conclude that the pre-Alpine tectonic history prior to accretion is more complicated than a single event would allow.

**TABLE 1 – LIST OF SAMPLES**

| Sample | Rock type and mineralogy | Sample location | | Deformation structure analysed | Ages |
|---|---|---|---|---|---|
| | | Lat (°N) | Long (°E) | | |
| Port Beach tectonic slice (structurally highest in Ios lower plate) | | | | | |
| IO18-01 | augengneiss:<br>quartz±garnet±potassium feldspar<br>±hornblende± white mica±biotite | 36°42.9'N | 25°17.2'N | First generation (pre-shear zone operation) white mica as porphyroclasts in matrix | 192 ± 1.3 Ma<br>188 ± 1.0 Ma<br>84.2 ± 2.3 Ma |
| | | | | K-feldspar crystals in groundmass | 592 ± 8.7 Ma<br>166 ± 2.8 Ma<br>39.8 ± 3.5 Ma |
| Mylopotas tectonic slice (structurally mid-level in Ios lower plate) | | | | | |
| IO17-03[*] | garnet-mica schist:<br>Quartz±garnet±biotite±rutile ±<br>white mica±potassium feldspar | 36°42.9'N | 25°17.2'E | White mica from south-directed shear zone deformation fabrics | 76.9 ± 0.7 Ma<br>36.0 ± 0.5 Ma |
| IO17-04 | garnet-mica schist:<br>Quartz±garnet±hornblende±biotite<br>±potassium feldspar±white mica | 36°42.9'N | 25°17.2'E | White mica from south-directed shear zone deformation fabrics | 163 ± 1.0 Ma<br>174 ± 1.1 Ma |
| IO17-05[*†] | garnet-mica schist:<br>Quartz±hornblende±garnet±biotite<br>±potassium feldspar±white mica | 36°42.9'N | 25°17.2'E | White mica from south-directed shear zone deformation fabrics | 81.0 ± 0.6 Ma<br>58.8 ± 1.5 Ma |
| IO18-05 [†] | garnet-mica schist:<br>Quartz±garnet±biotite±rutile ±<br>white mica±potassium feldspar | 36°42.5'N | 25°17.2'E | White mica from south-directed shear zone deformation fabrics, sample represents the structurally lowest level of this tectonic unit | 50.7 ± 0.4 Ma<br>43.8 ± 1.1 Ma |
| Augengneiss basement (structurally lowest in Ios lower plate) | | | | | |
| AG03-03 | Re-analysis on Ar/Ar geochronology data produced and published in Forster and Lister, 2009. | 36°42.2'N | 25°17.2'E | White mica from south-directed shear zone fabrics overprinted by north-directed shear zone | 73.9 ± 0.6 Ma<br>70.4 ± 0.8 Ma |
| | | | | Groundmass and porphyroclast k-feldspar grains subjected to deformation by two shear zones | 84.8 ± 0.7 Ma<br>~ 13 ± 0.1 Ma |
| AG03-05 | augengneiss:<br>Quartz± biotite± hornblende±<br>white mica±potassium feldspar | 36°42.2'N | 25°17.3'E | White mica from south-directed shear zone fabrics overprinted by north-directed shear zone | 72 ± 0.6 Ma<br>68.3 ± 0.3 Ma |

Samples stored in collections of the Structure Tectonics Team, Research School of Earth Sciences, Australian National University, Canberra, 2601 Australia

[*] Rock sample preserved evidences of eclogite facies after the first (deformation) fabric

[†] The younger age from the less retentive argon diffusion domain in the grain analyzed is comparable with the $\Delta_{1B}$ eclogite event in Syros

## 3.1 Port Beach tectonic slice: the structurally shallowest level

The Port Beach tectonic slice just beneath the Ios detachment represents the structurally shallowest level of the Ios basement terrane and is made up of two lithological units: a thin slice of structurally above garnet-mica schist and the underplating augengneiss with quartz porphyroclasts. Both units preserved numerous recumbently folded veins, isoclinal folds and boudinage structures overprinted by the south-directed SCSZ. A section of altered, greenschist facies garnet-mica schist with chloritoid replacing garnets was observed in the garnet-mica schist tectonic slice near the tectonic contact between Ios upper and lower plate. Beneath the altered zone, garnet porphyroblasts in the garnet-mica schist overgrew a pervasive white mica fabrics, and were rotated to form and δ-type clasts during SCSZ operation (Fig. 6a, cf Passchier and Simpson, 1986).

Two generations of white mica were observed in this unit, including pre-mylonite porphyroblasts (now present as muscovite fish with dynamically recrystallised rims) and the younger, recrystallised phengite (separated into wm2 and wm3 based on their overprinting relations) that intergrew with dynamically recrystallised K-feldspar and quartz (Fig. 6b). Silicate content of the phengite deformation fabric is ~3.40–3.45 Si a.p.f.u. (Fig. 6c). This, along with the mineral assemblage of quartz ± garnet ± potassium feldspar ± hornblende ± white mica (phengite) ± biotite, suggests P-T conditions of 1.8–2.2 GPa and 500-600ºC based on calculations in Massonne and Schreyer (1987), Patrick (1995) Velde (1967) and Kamzolkin et al. (2016). The small garnet blasts are preserved in low-strain zones, particularly adjacent to pull-aparts marked by quartz filled voids (Fig. 6b). Amphibolite facies may have taken place in Hercynian time, but during the Late Cretaceous it appears that the Port Beach tectonic slice was subjected to high-pressure eclogite facies conditions .

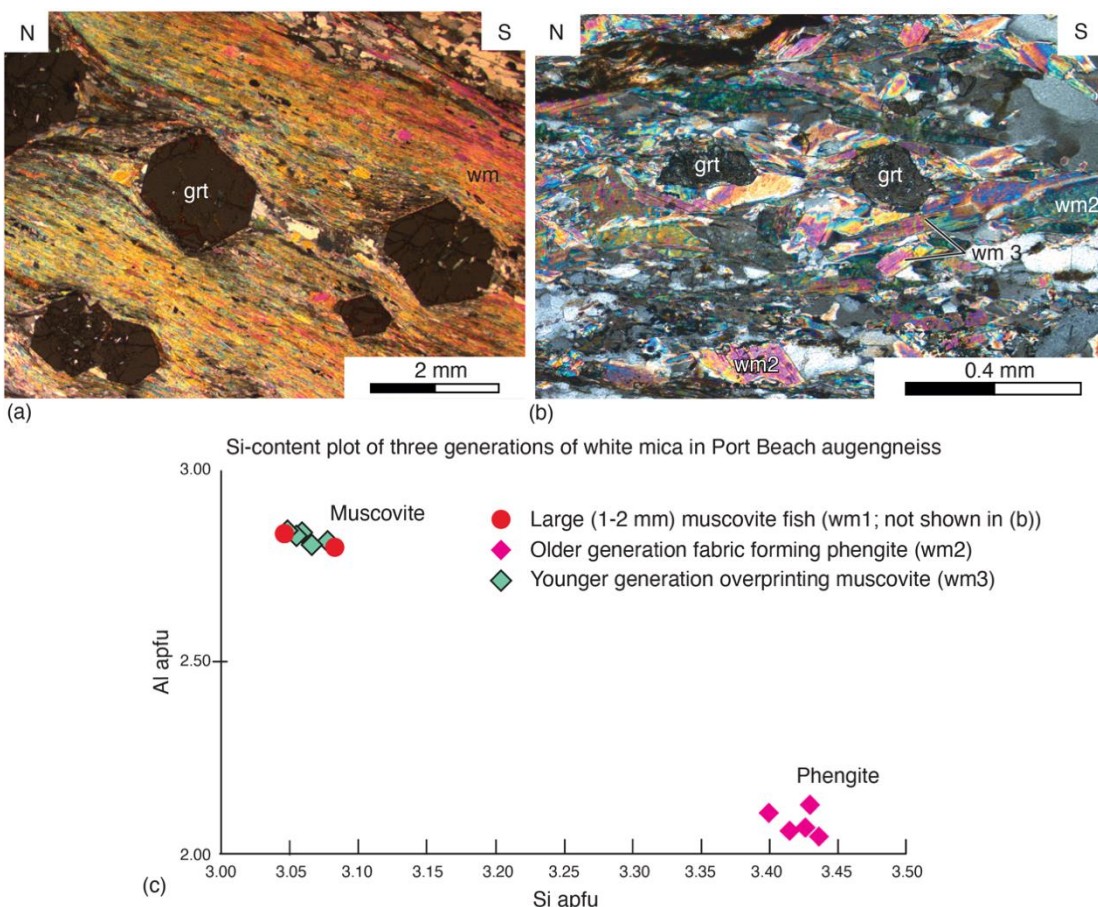

Figure 6: Microstructures analysis in the Port Beach tectonic slice. (a) garnet porphyroblasts with δ-type pressure shadows in the Port Beach garnet-mica schist (IO18-04). (b) two generations of overprinted white mica deformation fabrics, with new-grown (wm3) layer-parallel phengite overprinting the 'lens' shaped (wm2) phengite which has its mineral cleavage oblique to the fabric. (c) the plot of Si-content illustrating the presence of phengite and muscovite in the Port Beach augengneiss (IO18-01).

## 3.2 The structurally mid-level garnet-mica schist tectonic slice

Field observations in this garnet-mica schist slice and in the top part of the underlying augengneiss core identified evidence for multiple alternating and overprinting deformation events such as recumbent folds overprinted by extensional shear zones. The effects of the (here) N-S striking SCSZ fabric is pervasive, and most of the early fabrics recrystallised during this extensional episode. Nevertheless, relicts of earlier fabric are observed in low strain zones besides large porphyroclasts.

Samples collected from the structurally deeper level of this tectonic slice in the north Mylopotas headland preserved the most complex mineral growth and micro-deformation history. We note that metabasite was observed sporadically in the augengneiss basement, with mineral assemblages that suggest it was subject to transitional greenschist-blueschist metamorphism during the $\Delta_{1D}$ event of Forster et al. (2020). The protolith for such metabasite pockets is likely to have been intermediate-mafic

intrusive dykes that folded and deformed with the country rock (see structural maps in the supplementary material). A late-developed, intense north-sense shear zone defines the structural contact between the garnet-mica schist and the augengneiss in this locality. Fluid associated haematite nodes are found in the top three metres of an intense shear zone at the contact between the juxtaposed garnet-mica schist tectonic slices and the augengneiss core.

The four samples presented in Table 1 have garnet porphyroclasts recording multiple mineral growth events and preserved earlier fabrics as inclusions. Sample IO17-03 and IO17-04 retained the earliest formed garnets (some of which are large, exceeding 2–3 cm in diameter). Sample IO17-03, IO17-04 and IO17-05 preserved different stages of micro-tectonic events during SCSZ operation. The larger, first generation (1-2 cm diameter) garnet porphyroblasts are intact in IO17-03, fragmented during shear zone operation in IO17-04 and acting as porphyroclasts during deformation in IO17-05. Relics of earlier fabrics

are preserved in the low strain zone behind garnets in IO17-03 and IO17-04 and are microstructurally distinguishable, whereas fabrics in sample IO17-05 is almost completely reset by the SCSZ. IO18-05 is collected from a fold hinge of a recumbent fold ($M_1$ folding) that is overprinted by the SCSZ ($D_2$ crustal stretching), it represents the structurally lowest level of this tectonic slice. Haematite nucleating on the deformation fabric is observed, and relics of earlier deformation fabrics with rutile are preserved as inclusions in the garnet porphyroblasts.


Mineral chemistries with the three generations of garnet growth recognised in sample IO17-03 are chemically similar to almandine (see supplementary material – electron microprobe analysis (EPMA)) but are iron enriched and calcium depleted with slightly higher magnesium content compared to end-member almandine. Dynamically recrystallized, south-sense white mica fabric wraps around the larger (2–3 cm diameter) garnet porphyroblasts, with second generation garnets growing over

this fabric. The two younger garnet growth events are close in time, producing crystals of different size: the 2–3 mm diameter

crystals with uniform colour and the 5–8 mm diameter garnets have a zoned mineral growth (Figs. 7a-b). Observation on the two types of crystals identify the light red (Ca-depleted, Mn-enriched) garnets as the first growth event, followed by the second growth event producing black (Mn-depleted, Ca-enriched) garnets (Fig. 7c; see supplementary material). Although the later greenschist facies is pervasive across the outcrop, with chlorite overprinting the south-directed white mica deformation fabric,

traces of rutile crystals 'floating' in the relicts of earlier (pre-SCSZ) fabric in the low strain zone are preserved in IO17-03 (Fig. 7d). This also implies a higher-pressure history (potentially eclogite facies) than previously recognised.

The earliest microstructure observed was within garnet porphyroblasts, rotated during shear zone operation. This could be inferred from the oblique angle between white mica–rutile inclusions (in IO18-05) and the recrystallised groundmass (Fig. 8a,

showing core of a garnet porphyroblast). As the inclusions were fine grained, energy dispersive X-ray spectroscopy (EDS) analysis was used to confirm the presence of rutile in the included fabric. Electron microprobe analysis (EPMA) identified the chemical composition the garnet porphyroblasts (5–8 mm diameter) as between almandine and grossular, nucleated on Al-rich white mica during operation of an early shear zone, and continuing to grow during deformation until they reached Al-depleted, foam-textured quartz in pressure shadows (supplementary material – EPMA analysis results). Upon reaching the foam-textured

quartz, the fluids then corroded grain-boundaries, allowing the new-grown garnet to develop a skeletal structure in which the original foam texture in the incorporated quartz grains can still be recognised (Fig. 8b, bottom right corner). Late-stage (first order) grey albite grew in exsolution trails preserved across the garnet porphyroblasts, implying decompression as the garnet porphyroclast fractured during shear zone operation. (Fig. 8b, supplementary material –EPMA analysis results).

Dynamic recrystallisation of white mica (phengite) and quartz in the groundmass of IO17-03 and IO18-05 occurred synchronously during shear zone operation. The Si-content of phengite in sample IO17-03 and IO18-05 suggests that the phengite grew under P-T condition up to 450-500°C (based on the presence of garnet and biotite) with pressure in the range 0.7–1.7 GPa based on calculations in Massonne and Schreyer (1987), Patrick (1995) Velde (1967) and Kamzolkin et al. (2016) (Fig. 8c). We therefore suggest that the structurally mid-level garnet-mica schist tectonic slice also recorded a complex history

of deformation and metamorphism with evidence of high-pressure transitional amphibolite–eclogite facies metamorphism.

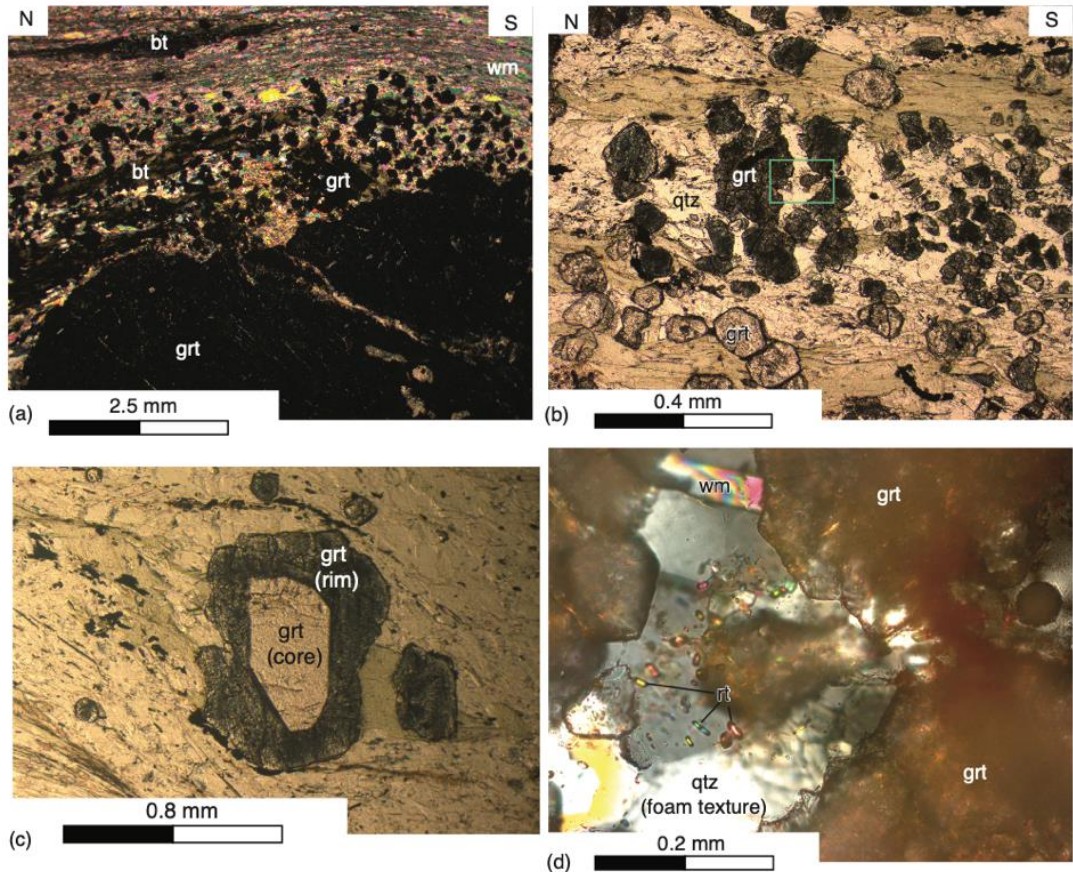

**Figure 7: Microstructure analysis in the garnet-mica schist tectonic slice (sample IO17-03). (a) White mica dominated deformation fabric with minor biotite relict of early fabric surrounds large 2–3 cm diameter garnets, with younger 2–3 mm garnets grown on the deformation fabric. (b) Thin section under plane polarized light: two types small, second generation garnets with different chemical compositions are identified (see supplementary material – EPMA analysis results). (c). A slightly larger (~ 4mm diameter) second generation garnet with a zoned crystalline texture. (d) magnified view of the green box in (b), a euhedral second-generation garnet that grew into a quartz foam texture with relict rutile floating in the void space.**

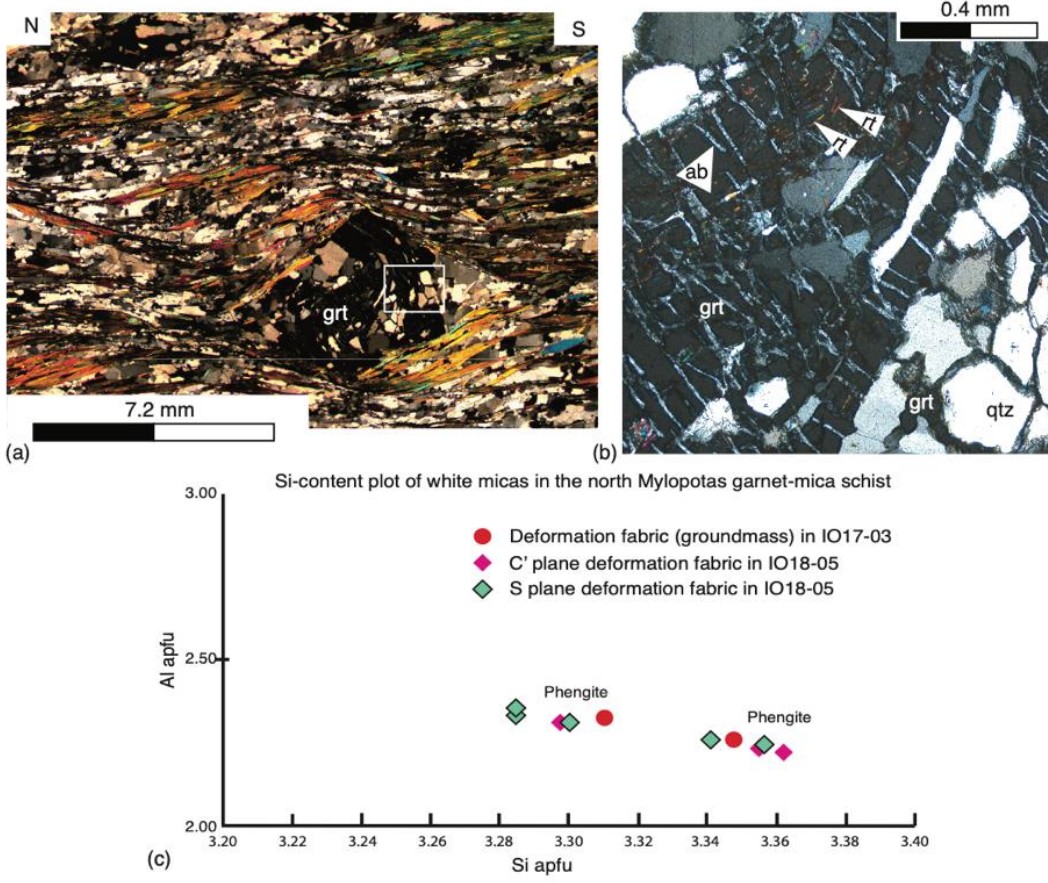

Figure 8: Sample IO18-05, garnet-mica schist collected in the fold hinge of the earliest fold (overprinted by the south-directed shear zone) in the mid-level garnet-mica schist tectonic slice. (a) a garnet porphyroblast preserving relicts of earlier deformation fabrics as inclusions and developed a skeletal structure once it reached the Al-depleted zone. (b) rutile inclusions and albite exsolution trails observed in the garnet porphyroblast. (c) a Si-content plot indicating the presence of two phengite groups in the garnet-mica schist.

### 3.3 The augengneiss core: the structurally lowest level

The structurally deepest augengneiss core of the Ios basement terrane is characterized by large (0.3-1.0 cm) K-feldspar xenocrysts preserved as porphyroclasts. Rocks in this locality were deformed by the south-directed South Cyclades Shear Zone (SCSZ) then variably overprinted by narrow north-directed shear zones. Occasionally, single hornblende porphyroclasts wrapped by a south-sense white-mica shear fabric could be observed (e.g., Fig. 9a, in a thin-section cut parallel to the stretching lineation). Pre-deformation hornblende was also observed in low-strain zones adjacent to these K-feldspar porphyroclasts (Fig. 9b). K-feldspar porphyroclasts surrounded by dynamically recrystallised white mica and quartz in these sample were fractured

by shearing, with recrystallisation at the edges (Fig. 9b). The youngest microstructures observed in these samples are quartz
filled cracks. The augengneiss basement records evidence of recumbent folding during crustal shortening, followed by ductile
stretching under a south-directed shear zone. All of this occurred before the augengneiss was juxtaposed against the garnet-
mica schist slice by an intense north-directed shear zone.

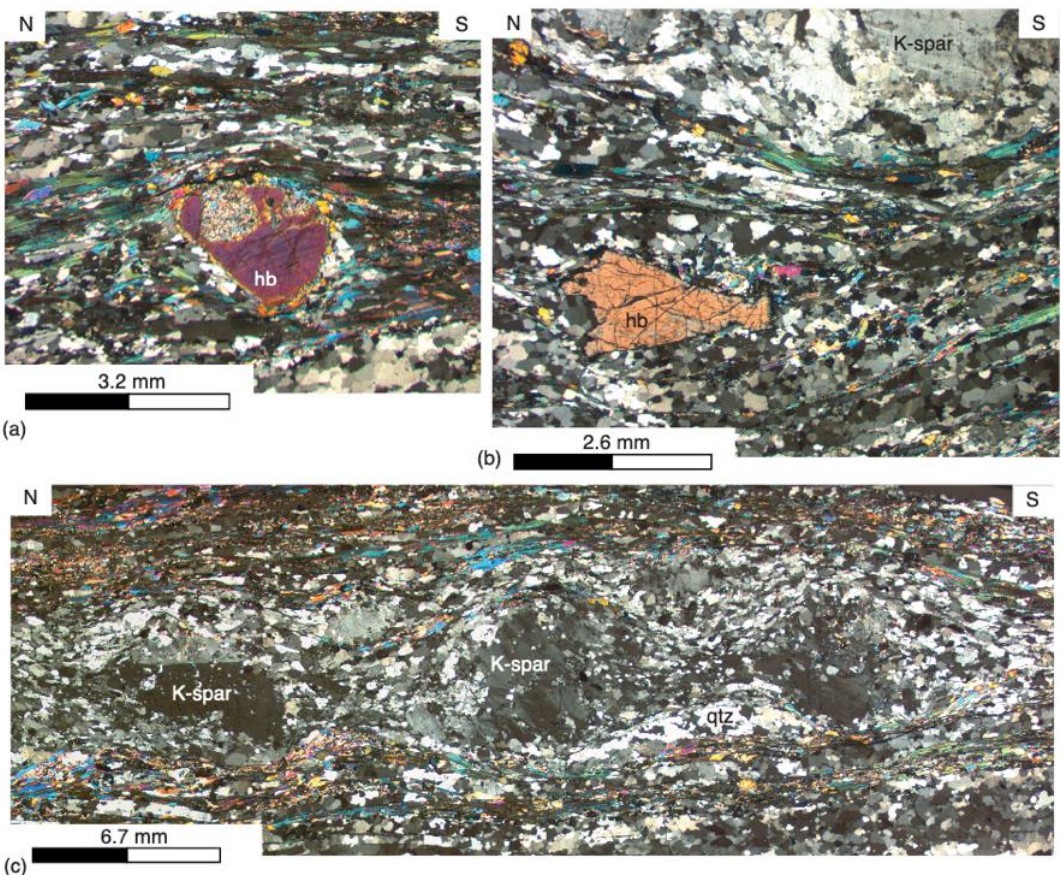

**Figure 9: Microstructures analysis on the south Mylopotas headland augengneiss (AG03-03, AG03-04, AG03-05). (a) an older
hornblende preserved as large porphyroclasts wrapped by younger, recrystallized white. (b) K-feldspar porphyroclasts with minor
recrystallisation limited to their boundaries, and a hornblende xenocryst preserved in a low-strain zone (AG03-04). (c) K-feldspars
porphyroclasts overprinted by both the earlier south-directed shear zone and the younger north-directed shear zone, forming 'micro
boudinage' structures (AG03-03).**

## 4 Argon geochronology

To provide time constraints on mineral growth events and deformation observed across the three tectonic slices, new $Ar^{40}/Ar^{39}$ geochronology data was collected using furnace-based step heating experiments conducted under ultra-high vacuum (UHV) conditions. These enabled new data that allowed recognition of Late Cretaceous Asteroussian ages in the garnet-mica schist mid-level unit where relicts of earlier, rutile containing fabrics are preserved. Argon geochronology was performed on white micas from deformation fabrics and on the K-feldspar porphyroclasts, with results summarised in Table 1. White micas with grain sizes ranging from 250μm to 420 μm were used as microstructural analysis identified the relicts of the earliest fabric to be of larger grain sizes compared to the dynamically recrystallised Alpine deformation fabrics (refer to supplementary material on $^{40}Ar/^{39}Ar$ analytical technique). No new analysis is performed on the two augengneiss samples AG03-03 and AG03-05 collected at the augengneiss core but the data previously published by Forster and Lister (2009) was re-examined in order to link microstructures observed with the reported Late Cretaceous dates.

The age spectra produced varied in their character depending on the structural character and rock type. For example, the morphologies of the argon spectra obtained from the garnet-mica schist are different and distinct in comparison with those obtained from the augengneiss. The phengitic white micas from the thick garnet-mica schist slice produced spectra with a characteristic 'hump-shaped' partial plateau, whereas age spectra from phengitic white mica in the underlying augengneiss generally produced spectra with a partial plateau rising to a peak in the final heating steps. The Late Cretaceous Asteroussia ages are always preserved in phengitic white mica, and since this appears to be highly retentive of radiogenic argon, these are likely to be growth ages and hence key to identifying older Asteroussian fabrics overprinted by younger Alpine events. Previous research suggested that the later-formed shear zones operated in this area operated in the Argon Partial Retention Zone (Baldwin and Lister, 1998; Forster and Lister, 2009), but this was on the basis that it had been assumed that all the white mica was muscovite, which is not correct. The complex age spectra preserve and record the effect of multiple deformation and metamorphic mineral growth events, but they are preserved only because phengitic white mica (especially under high pressure conditions) is extremely retentive of argon (Lister and Baldwin, 1996; Warren et al., 2012).

Argon geochronology on white mica and k-feldspar grain separates from the three tectonic slices in the Ios basement terrane yielded age clusters in Early-Middle Jurassic, Late Cretaceous, Eocene–Oligocene and Oligocene–Miocene time (Table 1). However, evidence for Jurassic and Cretaceous ages is exclusively restricted to argon populations retained in phengite, or, in the case of IO18-01, to the large muscovite fish. All white mica analysed yielded Arrhenius plots that unequivocally demonstrate both phengite and muscovite components, e.g., IO17-05 in the garnet-mica schist, and AG03-03 in the augengneiss (Figs. 10-11; see corresponding figures in the supplementary material). The phengitic components produce significantly high activation energy estimates, in the range 103–115 kcal/mol (431–481 kJ/mol) compared to estimates from the muscovite domain, in the range 54–61kcal/mol (226–255 kJ/mol) (Fig. 12; see corresponding figures in the supplementary data). The estimated retentivity of the phengite implies that the ages measured are growth ages, since metamorphic temperatures were less than the inferred closure temperatures from the Arrhenius plots. Therefore, it appears that we have been successful in being able to directly date microstructures produced during the Late Cretaceous Asteroussia event.

The garnet-mica schists that produced the Late Cretaceous Asteroussia ages were collected in the northern headland of the Mylopotas Beach, Ios (Fig. 5). We have already noted that microstructural analysis of the garnet-mica schist IO17-03 and IO17-05 demonstrated multiple episodes of white mica growth. The older grains in the deformation fabric are 180 µm to 450 µm in diameter, whereas the younger grains developed during or after later shear zone operation are elongate with dimensions range 50 µm to 90 µm. Note that the older generation phengitic white micas (355-450 µm grains) were tediously hand-picked for this sample. The $^{40}Ar/^{39}Ar$ results suggest several different gas populations retained in the crystal lattice, with a younger gas population in the less retentive domain and an older gas population in the more retentive domain that dominated gas release. The older argon population accounted for 90% argon released in IO17-03 white mica and created a partial plateau ('hump') with peak minimum age of 76.9 ± 0.7 Ma in the phengitic part of the age spectrum (Fig. 10a; see corresponding figures in supplementary data). The younger gas population that accounts for the 5% of initial argon release comes from muscovite formed later in the geological history, during operation of the SCSZ. However white mica from the relicts of earlier fabrics in IO17-05 preserved an older argon population with peak minimum age of 81 ± 0.6 Ma. A younger gas population in the less

retentive domain of the earlier white mica fabric in IO17-05 record an age of 59 ± 1.5 Ma, which is comparable with estimates

for the timing of the $\Delta_{1A}$ and $\Delta_{1B}$ Alpine events (Forster et al., 2015; Huet et al., 2009).

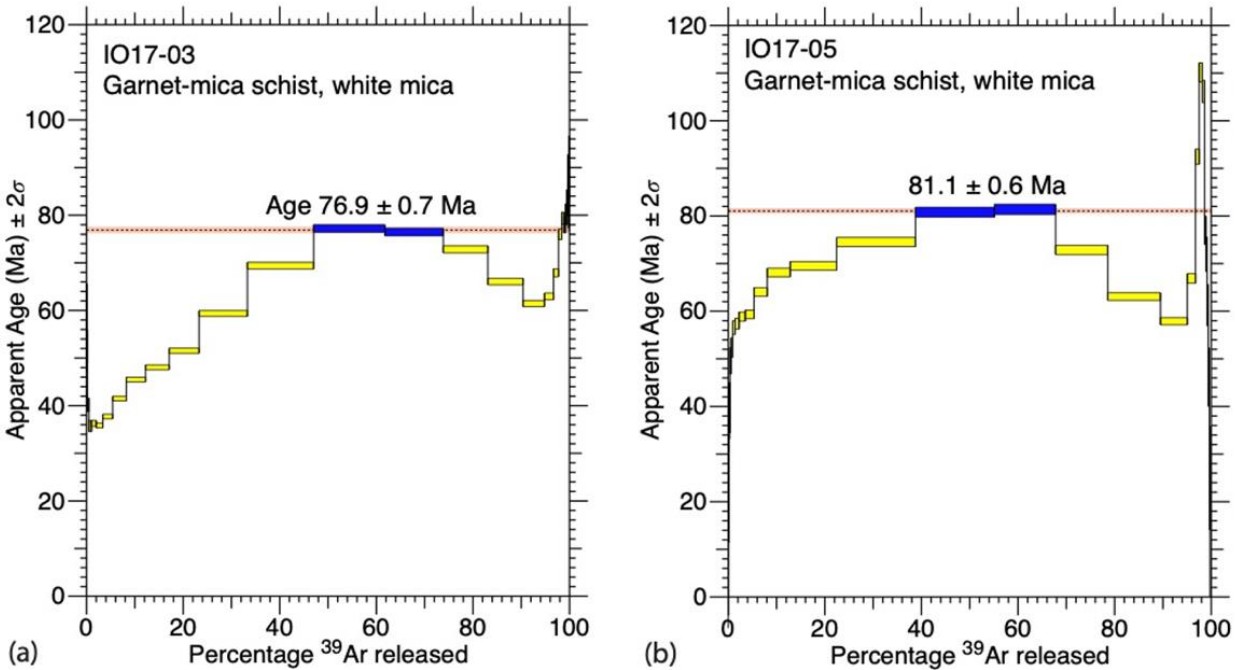

Figure 10: White mica age spectra from the structurally mid-level garnet-mica schist unit (IO17-03, IO17-05) produced Late
Cretaceous ages, from mica grown during the Asteroussia event.

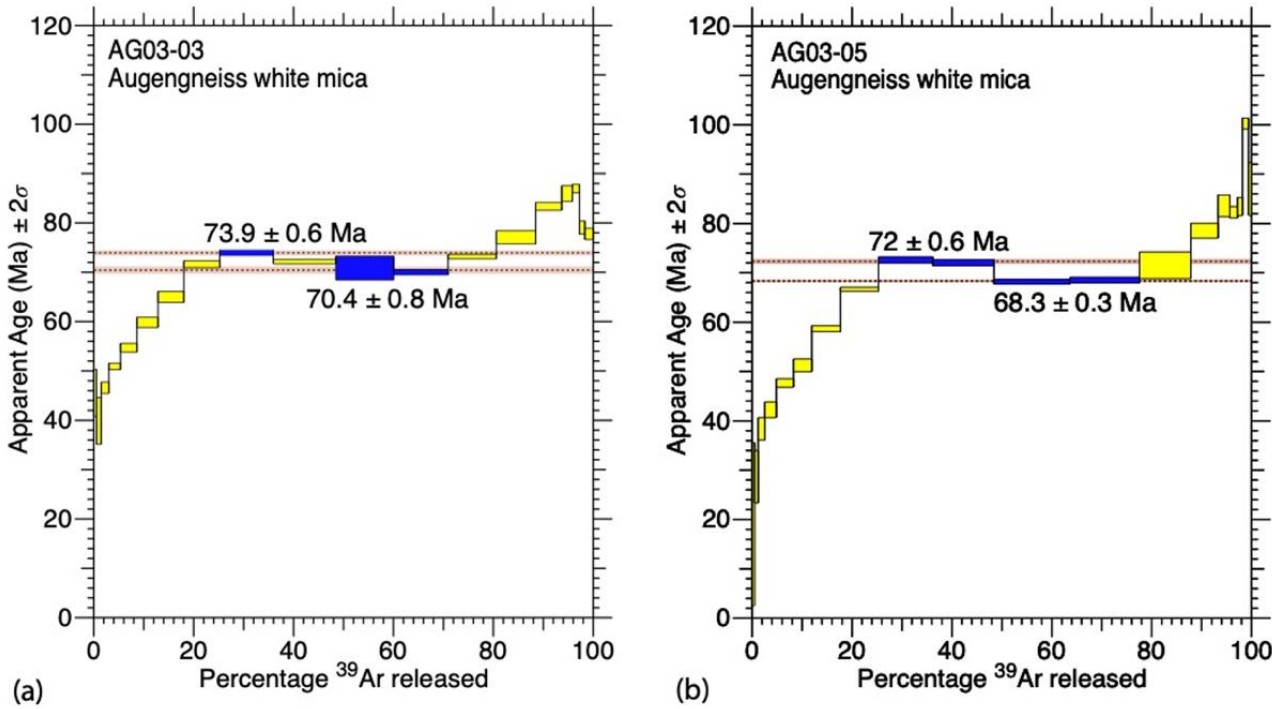

**Figure 11: White mica age spectra from the structurally lowest augengneiss core (AG03-03, AG03-05). The complex age spectra are a result of multiple argon populations degassing at different temperature during the step-heating experiment. The augengneiss basement was subjected to multiple deformation events, but a significant argon population is derived from phengite with Late Cretaceous ages preserved. The argon geochronology data was published in Forster and Lister, 2009 and re-analysed in this study.**

Samples AG03-03 and AG03-05 were collected from the augengneiss core, and $^{40}$Ar/$^{39}$Ar geochronology on isolated white mica deformation fabrics were performed in the study reported in Forster and Lister (2009). The two rocks are microstructurally similar, fabrics underwent minor recrystallisation during south-directed then north-directed shear zone operation. We reanalyzed the diffusion experiment result in this study as white mica grain separates from the two samples produced Late Cretaceous dates (Forster and Lister, 2009). Application of the method of asymptotes and limits on the AG03-03 white mica age spectrum yielded a range of ages from 70.4–74.0 Ma (Fig. 11a) for phengitic white mica (Fig 11a, see corresponding supplementary material). White mica deformation fabrics in augengneiss AG03-05 show an upper limit at 72 ± 0.6 Ma and a lower limit at 68.3 ± 0.3 Ma in gas release of the more retentive domain, representing the minimum and maximum ages of a single Late Cretaceous event respectively (Fig. 11b). The older ages in the age spectra may represent even older relict fabrics.

From these data it is evident that the structurally mid-level garnet-mica schist and the underlying augengneiss basement were subjected to complex deformation history with multiple events occurring from Early Cretaceous to Miocene time.

White mica and K-feldspar differ in their $^{40}Ar/^{39}Ar$ systematics and grow and respond differently to deformation. K-feldspar grain separates were collected from all augengneiss samples in an unsuccessful attempt to pinpoint the microstructure(s) responsible for the Late Cretaceous date reported in the white mica. Forster et al. (2014) reported that K-feldspars required analysis with isothermal steps so as to recognise contamination at each temperature increase in the step heating procedure (i.e., isothermal steps being two or more heating steps at the same temperature). The first step is referred as a cleaning step and is

not included in the interpretation of the spectrum. This same methodology is used on the K-feldspar analysis in this study.

In AG03-03, Forster and Lister (2009) observed larger K-feldspars (porphyroclasts; 2000 – 6500 µm) and small K-feldspar grains (500-700 µm) interspersed between aligned white mica grains that recrystallised during later deformation. Step-heating experiments on the K-feldspar grain separates (including both porphyroclasts and small grains) from AG03-03 produced

saddle-shaped apparent age spectrum with a lower limit at ~13 ± 0.1 Ma (Fig. 12a). The last argon release steps produced a peak at 84.5 ± 0.8 Ma, comparable to the date obtained from white mica from the same sample (Fig. 11a). The Arrhenius plot of K-feldspar in sample AG03-03 shows two distinct argon diffusion domains (Fig. 12b). This suggests that the K-feldspar in the south Mylopotas augengneiss also preserved complex deformation history, with the oldest (and most retentive domains) regrown during the Late Cretaceous event. The Arrhenius data (Fig. 12b) shows that these older domains were capable of

retaining argon at temperatures well above those recorded by the metamorphic assemblages, implying that these are growth ages, requiring the original potassium feldspar to have been replaced during metamorphism and/or metasomatism by this time.

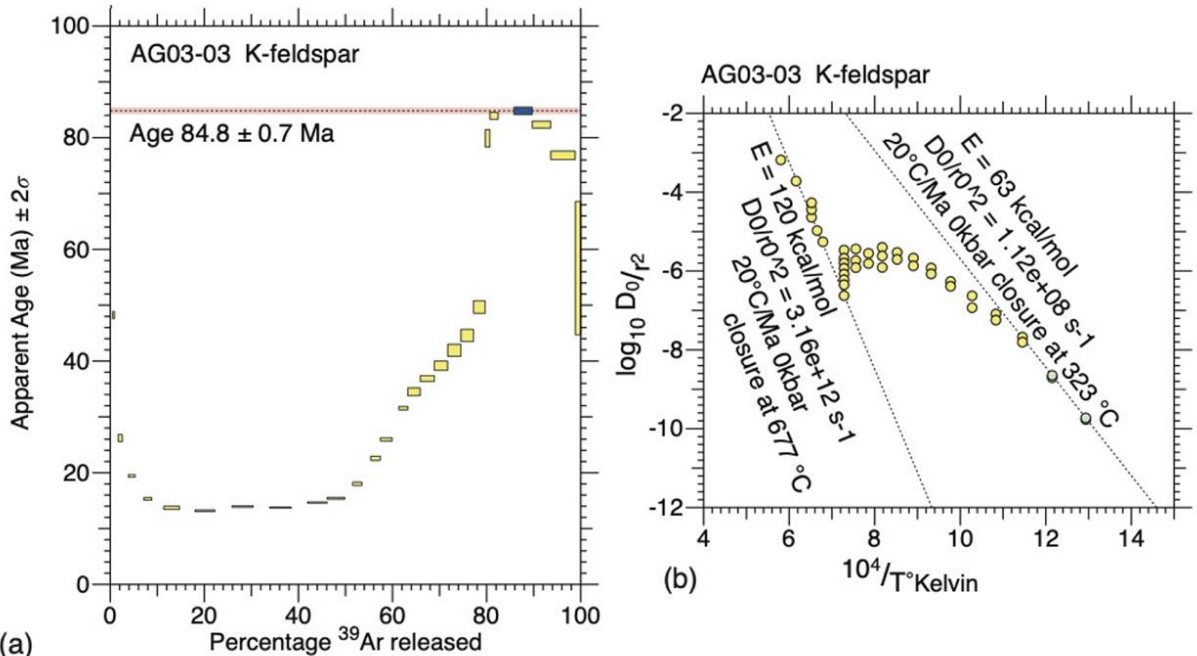

**Figure 12: a) a K-feldspar age spectrum from the structurally lowest augengneiss core (with isothermal cleaning steps removed). The origin of the younger part of the age spectrum is discussed by Forster and Lister (2009). The Late Cretaceous age is preserved in the retentive core domains. (b) the corresponding Arrhenius plot shows two diffusion domains with significantly different activation energies.**

## 5 Discussion

### 5.1 Evidence of the Asteroussia event in the Ios lower plate

Microstructurally, our study has conclusively identified the presence of more retentive phengite in a fabric that was later overprinted by dynamically recrystallized white mica and quartz. The earlier metamorphic fabrics formed under conditions that potentially reached eclogite facies. Our UHV $^{39}$Ar diffusion experiments show that this phengite is highly retentive, allowing preservation of the growth ages of the white mica that formed during these earlier events. Thus, despite intense overprinting during Alpine deformation events, the Late Cretaceous argon populations were retained. This is consistent with the concept of an Argon Partial Retention Zone in which mineral grains undergo some partial resetting by diffusion, but where recrystallisation causes the most effects (Baldwin and Lister, 1998). However, the concept of a partial retention zone is appropriate only for systems with single diffusion domains and no variation of activation energy, which is not the case here.

Identification of the Late Cretaceous age in the Ios lower basement has been interpreted as a result of mixing (e.g., Andriessen et al., 1987): in other words, defining these dates as "intermediate" ages due to excess radiogenic argon or simultaneous degassing of the Alpine mica and the older Hercynian micas. However, here we have shown that these ages represent a period of Late Cretaceous deformation and metamorphism. Therefore, the Ios basement may indeed be part of the Asteroussia terrane. However, pressure-temperature estimates from phengites in the Ios lower plate record high pressure conditions, contrary to what has been observed in Asteroussia klippen across the Cyclades, albeit preserved at different structural levels. This suggests that more than one set of tectonic slices may have preserved the Asteroussian ages, and we have already pointed to the role that tectonic shuffling may play in producing such variation. It is important in this aspect that the Ios data is the first report of Asteroussia ages in a terrane of unmistakeably Gondwanan affinity (Keay and Lister, 2002).

There may be an earlier Hercynian history: the earliest reported argon age in Ios is a single K/Ar hornblende date reported to be post-Hercynian (268 ± 27 Ma) by Andriessen et al. (1987) and Flansburg et al. (2019). However, based on the peak metamorphic P–T conditions documented across the Cyclades, it may be that the Asteroussian terrane slices record a variety of metamorphic pressure conditions (Table 2, and reference therein). Rocks from the basement slices on Ios suggest the occurrence of high pressure–medium temperature conditions based on the microchemistry preserved in relicts of earlier deformation fabrics.

# TABLE 2

## PEAK METAMORPHIC CONDITION OF THE ASTEROUSSIA EVENT ACROSS THE CYCLADES

| Island | Published studies | Sample details/ methodology | Peak metamorphic condition |
|---|---|---|---|
| **Tinos** | Patzak et al., 1994 (as cited in Be'eri-Shlevin et al., 2009) | Interlayered amphibolite–paragneiss sequence in Akrotiri unit | 650–750 MPa 530–610 ℃ |
| **Donoussa** | Kolodner et al., 1998 | P-T estimates on garnet-sillimanite-biotite-quartz assemblage observed in pelitic rocks. | Core of garnet 400–500 MPa 600–650 ℃ |
| | | Distinct chemical zoning of garnets allowed P-T calculation in core and rim respectively | Rim of garnet 250–350 MPa 550–580 ℃ |
| **Anafi** | Be'eri-Shlevin 2009 | EPMA analyses of garnet–biotite pairs from garnet-biotite paragneiss sample that occur as thin (1–2 m thick) layers within the structurally intermediate level of the Asteroussia Unit. Garnet–biotite temperatures were calculated using the equation of Ferry and Spear (1978). | Core of garnet & biotite ~720± 50–740± 50 ℃ 200–600 MPa Rims of garnet & biotite 634± 50–650± 50 ℃ [*] 200–600 MPa |
| | Be'eri-Shlevin 2009 (cont.) | Sample collected from a massive amphibolite exposure in the structurally | |

|  |  | intermediate level of the Asteroussia Unit |  |
|---|---|---|---|
|  |  | edenite–tremolite (ed–tr) reactions | 677–726 ºC |
|  |  |  | 200–600 MPa |
|  |  | edenite–richterite (ed–ri) reactions | 605–643 ºC |
|  |  |  | 200–600 MPa |
| **Crete** | Seidel 1981 | Peak metamorphism P-T conditions estimated from critical mineral assemblage of the outcrop of a variegated series consisting of: tholeiitic ortho-amphibolites, para-amphibolites, andalusite and sillimanite-cordierite-garnet bearing mica schists, calcsilicate rocks, and marbles. | 400–500 MPa maximum temperature ~ 700 ºC |
|  | Anderson and Smith,1995 (as cited in Langosch et al., 2000) | Al-in- hornblende barometer Granodiorites of eastern Crete | 100–200 MPa Maximum temperature = 700 °C |
|  |  | Granites and granodiorites of central Crete | 250–400 MPa Maximum temperature = 700 °C |

| | | | |
|---|---|---|---|
| **Crete (cont.)** | Koepke and Seidel, 1984 (as cited in Langosch et al., 2000) | Peak metamorphism P-T conditions estimated from metamorphic assemblages of quartz – plagioclase – K-feldspar – sillimanite – biotite – garnet – cordierite in pelitic paragneisses at central Crete | upper amphibolite facies: 400–600 MPa 650–700 °C |
| | Langosch, 1999 (As cited in Langosch et al., 2000) | Calculated by thermobarometric calibrations of Bhattacharya et al. (1988, 1992), Dwivedi et al. (1998), Koziol and Newton (1988) and Holland and Blundy (1994) Peak metamorphism P-T conditions estimated from metamorphic assemblages of | |
| | | (1) quartz – muscovite – chlorite – garnet – andalusite – plagioclase and | 680–730 °C 500–600 MPa |
| | | (2) quartz – muscovite – biotite – staurolite – andalusite – plagioclase observed in metapelites of Asteroussian tectonic slices | lower amphibolite facies: ~ 550 °C 300 MPa |

## 5.2 Tectonic implications

The nature of the tectonic processes that affected the evolution of the terranes accreted by the Hellenic subduction zone remains controversial, e.g., comparing the papers by Forster and Lister (2009), Forster et al. (2020) to that written by Huet et al. (2009) and Huet et al. (2011). However, the polemic seems misguided. The architecture of Tethyan orogenic belts, the Hellenides included, invariably involves a nappe- or a terrane-stack, and all terrane stacks are created by thrusting. However, most if not all terrane stacks are also modified by later episodes of extension (e.g., as in Forster and Lister, 2009) leading to tectonic

shuffling. It is no different in the Cyclades. The Cycladic archipelago preserves the results of the destruction of an extensive terrane-stack that extended from the Hellenides in Greece to the Taurus Mountains in Turkey (Gautier and Brun, 1994a, b; Kempler and Garfunkel, 1994; McKenzie, 1977; Taymaz et al., 1991). The debate as to the nature of exhumation processes will not be resolved by a sole focus on the Cycladic eclogite-blueschist belt, as demonstrated in this paper.

The key questions surround the evolution of the terrane stack overall, rather than the details of the exhumation of an individual tectonic slice. The extrusion wedge (or forcible eduction) model suggests constant compression, resulting in the squeezing of softer material, so that it is extruded to the surface (Forster and Lister, 2008; Xypolias and Koukouvelas, 2001). The competing hypothesis, known as the tectonic mode switch or tectonic shuffle zone model, considers that thrust slices are exhumed by periods of crustal extension that take place in between episodes of crustal shortening caused by individual accretion events

(Forster and Lister, 2009). Dispute arises because of the focus on the exhumation of the Cycladic eclogite-blueschist terranes, whereas the continuing nature of the orogenic process means that (without question) the subduction megathrust had to have episodically leapt southward every time a new terrane was accreted (e.g. Lister et al., 2001; Ring et al., 2007; Huet et al., 2009; Forster and Lister, 2009). As the African plate migrated northward, terranes were first subducted, then sliced from the subducting lithosphere by the advancing subduction megathrust, and thus accreted to the terrane stack (e.g., Lister et al.,2001;

Lister and Forster, 2009).

For Ios, the question is how rollback of the subducting slab was able to throw the over-riding terrane-stack into horizontal extension immediately after the accretion of the Cycladic blueschist onto the Gondwanan basement from which the

Asteroussian terranes were derived (Fig. 4), in particular given the requirement thereafter of a massive southwards leap of the

outcrop of the active subduction megathrust. Previous work (e.g., Forster et al., 2020) has suggested that the Cycladic

blueschist belt had already been largely exhumed before it was thrust over the Ios basement terrane in Late Eocene time (from

~38 Ma, Fig. 4a). A first period of extensional tectonism formed the Ios metamorphic core complex, and this had commenced

by ~35 Ma, accelerating by the time of the Eocene–Oligocene transition. A second period of extensional tectonism then ensued,

after the Oligocene–Miocene transition, with extreme lithospheric extension triggering a major magmatic event, with intrusions

in and through the core of younger metamorphic core complexes across the Cyclades.

The Ios basement has been argued to be autochthonous, moving with Africa, and part of Gondwana (Flansburg et al., 2019;

Keay et al., 2001; Keay and Lister, 2002). Its accretion to the terrane stack is therefore likely to have been an event with

considerable tectonic significance. The magnitude of the southward leap of the subduction megathrust is thus unlikely to have

been accomplished without the development of a new lithosphere-scale structure. There are two end-member options: one

requiring that the slab peels free from the subduction megathrust (Fig. 3, using the slab peel hypothesis discussed by Brun and

Faccenna, 2008) while the other requires a subduction jump and slab breakoff (Fig. 4, cf. von Blanckenburg and Davies, 1995).

Although the slab-peel model is consistent with enhanced heat flow during crustal stretching after the accretion event, such a

model requires the asthenosphere to be exhumed to such shallow levels as to require significant partial melting of the uplifting

asthenosphere, which would a period of widespread basaltic volcanism, with volumes comparable to those observed in some

large igneous provinces. Such effects were not observed in the Cyclades. Sizova et al. (2019) also showed the "peel off" model

(Brun and Faccenna, 2008) to be unlikely in the Aegean region.

An alternative model involving slab necking (or boudinage) and break off must therefore be considered (e.g., Fig. 4). This

(provisional) three-staged 'slab break off' model more accurate describes Aegean tectonics by addressing how the terrane

stack was subjected to overall stretching with some evidence of melting such as plutonic intrusions in the centre of

metamorphic core complexes. This model also requires significant magmatism, but in consequence of fluids rising from a

devolatising slab which would lead first to crustal magmatism, such as the I-type granite of Ios, and later to the appearance of

arc volcanoes, as on Thera. Possibly the necking and eventual break off of the subducting slab and formation of a new subduction zone (Fig. 4d) might be of sufficiently small scale to escape observation in models based on P-wave tomography.

## 5.3 Unresolved issues

We do not understand why the Asteroussia event is recorded in the top-most slices of the terrane stack outcropped in other Cycladic islands, but is found only in the lower slice in Ios. Such architecture implies that the Cycladic eclogite-blueschist tectonics slices are 'sandwiched' between tectonic slices affected by the Asteroussia event, whereas on Crete the Asteroussia units are juxtaposed above the Vatos unit, the Arvi unit, the Pindos unit and the Tripolitza unit (e.g., Bonneau, 1984; Flansburg et al., 2019; Kneuker et al., 2015; Langosch et al., 2000; Martha et al., 2017; Martha et al., 2016, Palamakumbura et al., 2013; Seidel et al., 1976; Zulauf et al., 2002). This must have occurred sometime between mid-Oligocene–early Miocene time. Laterally, the unit is connected to the eastern Alps in the west and the Lycian ophiolite nappes, the Menderes Massif and the Sakarya Zone in Turkey (van Hinsbergen et al., 2020). Further work is required to validate the tectonic-shuffling hypothesis which is capable of explaining these observations.

Some authors suggest that tectonic slices outcropping on islands in the northwest (Andros, Tinos, Syros) are different to those on other islands such as Anafi, Nikoria, Donoussa, Ikaria and Crete (Altherr et al., 1994; Langosch et al., 2000; Martha et al., 2016). Arguments arise due to the difference in dates obtained (despite all being Late Cretaceous) and different results for geothermobarometry across islands with different lithologies and metamorphic facies (Kolodner et al., 1998; Langosch et al., 2000; Patzak et al.,1994; Seidel et al., 1976; Seidel et al.,1981; Yeung, 2019). Research in the upper and middle tectonic units in Tinos produced dates at 90-100 Ma and a peak metamorphic P-T estimate of 120 MPa at 450-500 °C (Avigad and Garfunkel, 1989; Avigad and Garfunkel, 1991; Bröcker and Franz, 1998; Patzak et al.,1994), whereas studies on Donoussa and Crete produced younger ages at 70-80 Ma and peak metamorphic P-T conditions at 300–600 MPa and 600-730°C (Be'eri-Shlevin et al., 2009; Keay and Lister,2002; Kolodner et al., 1998; Langosch et al., 2000; Seidel et al., 1976).

Our study reports metamorphic conditions with higher pressure, despite producing similar dates. Although the presence of phengite is wide-spread across the Ios lower plate, the highest pressures are inferred only in the garnet-mica schist unit and the

Port Beach tectonic slice. With no evidence of higher pressures in the underlying augengneiss unit, it is possible that a more complex deformation and metamorphic history has been recorded in these intermediate slices in the Ios terrane stack. These observations also reflect on a possible distinction between European and Gondwanan terranes, with evidence mostly preserved in the Alps and in the Pelagonian zone of Greece (Pourteau et al., 2013; Porkoláb et al., 2019; Regis et al., 2014; Thöni, 2006). Brown et al. (2014) reports evidence of Late Cretaceous intracontinental shear zone deformation across Africa, thus demonstrating that the ~70-80 Ma age is not limited to northern Tethys. Detrital zircon (DZ) analysis on pre-plutonic metasedimentary rocks in Ios lower plate by Flansburg et al. (2019) pushes tectono-magmatic histories of the southern Cyclades further in time to early Cenozoic. They noted a striking resemblance between their DZ age spectra from Ios lower plate to exposures on Crete, northern and central Peloponnese, the northern Hellenides and the siliciclastic cover sequence of the Menderes massif in western Turkey (Flansburg et al., 2019). Comparing these Ios DZ age spectra to those from northeast Africa and Arabia, they confirmed that the Cycladic basement terrane (outcropped in Ios lower plate) have a distinct peri-Gondwanan affinity (Flansburg et al., 2019). This led them to propose a tectonic model where the terrane was located along the northern margin of Gondwana in early Paleozoic and experienced pluton emplacement between ~335 and ~305 Ma in an arc setting (Flansburg et al., 2019).

Tectonic reconstructions by van Hinsbergen et al. (2020) demonstrated that major continental-scale events occurred across Eurasia and Gondwana from Late Jurassic–Late Cretaceous time. These global tectonic events involve continental-scale deformation such as the formation of the Alpine Tethys with microcontinents tearing from the south coast of Europe. In their reconstruction model, the Ios basement, along with other tectonic units in the Cycladic islands and Crete, are all part of a subducted Greater Adria continental ribbon. While our island-scale study cannot contribute to the discussion on whether Greater Adria was a single continental landmass or made up of several large islands, it is evident that Late Cretaceous deformation is wide-spread in both the European and Gondwana terranes.

Distinguishing European versus Gondwanan terranes in the Central Aegean and greater Mediterranean area will remain difficult. One central argument is the number of oceans present in the 'greater Tethys seaway' between Europe, Africa and

potentially Adria at Mesozoic and associated tectonic evolution (e.g., Channell and Kozur,1997; Kilias et al., 2010; Robertson et al., 2013). This argument mainly concerns the paleogeography of the Pelagonian unit outcropped in mainland Greece. It is thus of interest that evidence for Late Cretaceous ages is reported from white mica deformation fabrics isolated from the northern end of the upper Pelagonian unit (e.g., Kilias et al., 2010; Robertson et al., 2013). The Pelagonian unit may have been a continental ribbon (or micro-continent) separating two Tethyan realms: the Vardar Ocean in the northeast and the Pindos (or even Cyclades) Ocean in the southwest (e.g., Channell and Kozur,1997; Robertson et al., 2013). Such models imply high pressure metamorphism in the Pelagonian unit as the result of the attempted subduction of the continental ribbon (Robertson et al., 2013). Other reconstructions consider the Pelagonian unit as the eastern-most unit of a continental Adria terrane, adjacent to a single north-eastern oceanic basin (the Vardar Ocean) e.g., (Bortolotti et al., 2013; Ferriere et al., 2012; Kilias et al., 2010; Palamakumbura et al., 2013). These researchers disagree with the concept of a distinct Pindos Ocean both in Triassic and in Jurassic time (Kilias et al., 2010).

**6 Conclusion**

Our study reports evidence of a Late Cretaceous Asteroussia event (70–80 Ma) in the originally Gondwanan lower plate of Ios. Accretion of the Asteroussia terrane is a major event in the Aegean tectonic history. This required a (250–300 km) southward jump of the subduction megathrust. Renewed rollback after the accretion event triggered Oligocene extension and facilitated the exhumation of the Asteroussia terrane within the core of the Ios metamorphic core complex.

**Data availability**

$^{40}$Ar/$^{39}$Ar geochronology results of two augengneiss samples (AG03-03, AG03-05) were published in Forster and Lister (2009) and re-examined in this study. All new data collected in this study and presented in this article are provided in text and in the Masters' thesis of Sonia Yeung submitted for her Masters' programme to the Research School of Earth Sciences, Australian National University.

**Team list**

Sonia Yeung[1], Marnie Forster[1], Emmanuel Skourtsos[2], Gordon Lister[3]

[1]Structure Tectonics Team, Research School of Earth Sciences, Australian National University, Canberra, 2601 Australia

[2]Section of Dynamic, Tectonic and Applied Geology, Department of Geology and Geoenvironment, National and Kapodistrian University of Athens, Athens 157 72, Greece

[3] W.H. Bryan Mining and Geology Research Centre, Sustainable Minerals Institute, University of Queensland, Australia

**Author contribution**

All authors contributed to the writing of the manuscript and its conceptualisation. The paper extends part of a Master's thesis by Sonia Yeung, supervised by Marnie Forster and Gordon Lister.

**Disclaimer**

The article includes a minor part of the Masters' Thesis of Sonia Yeung submitted for her Masters' programme to the Research School of Earth Sciences, Australian National University.

**Conflicts of Interest**

The authors declare that they have no conflicts of interest.

**Funding Statement**

Research support were provided by the Australian Research Council Discovery Project [grants numbers: DP120103554 "A unified model for the closure dynamics of ancient Tethys constrained by Geodesy, Structural Geology, Argon Geochronology and Tectonic Reconstruction" and LP130100134 "Where to find giant porphyry and epithermal gold and copper deposits"]. Sample Irradiations were paid by the Research School of Earth Science argon facility, Australian National University and facilitated by the University of California Davis McClellan Nuclear Research Centre, CA, US.

**Acknowledgments**

The authors acknowledge the microprobe mineral chemistry analysis was performed in the facilities, with the scientific and technical assistance in Microscopy Australia at the Centre of Advanced Microscopy, The Australian National University. Sample Irradiations for $^{40}Ar/^{39}Ar$ geochronology were facilitated by the University of California Davis McClellan Nuclear Research Centre, CA, US. Argon analyses and microstructure analyses were performed at the Research School of Earth Sciences Laboratories at the Australian National University. Davood Vasegh in the Argon Lab provided technical assistance for the step heating experiments, Shane Paxton in the mineral separation facility provided technical assistance for sample preparation. Step-heating experimental results for $^{40}Ar/^{39}Ar$ geochronology were analysed using programmes *eArgon* and *MacArgon* developed by G.S. Lister (http://rses. anu.edu.au/tectonics/programs/).

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
