# Peer review of "Evidence for the Late Cretaceous Asteroussia event in the Gondwanan Ios basement terranes"

_Solid Earth, 2020_

## Referee Comment (RC1) · Franz Neubauer (Referee) · 9 Jan 2021

General remarks:

This potentially an interesting manuscript showing some evidence on a Late Cretaceous Asteroussian metamorphic event on Ios basement and correlate this with a megathrust of an Asterroussia nappe extending to Crete. Basically, this would be an interesting story for the international readership. On the way to be convincing with the data on which the story is based several additions should be added to the revised version. These include: (1) The microfabrics of dated samples are complex, and the history is based mostly on white mica generations. No EPMA data are given for the white mica and other critical minerals like the two-stage zoned garnet. Add this sort

of data as well as BSE images to show the distinction of fabrics. Garnet would be a prime goal to extract further detailed information on fabric and, in conjunction with other minerals, for P-T calculation. (2) Add detailed information on the generalized P-T conditions for the Late Cretaceous Asteroussia event, on which the HP event is based except simply the phengite composition. (3) White mica wm2 and wm3 have similar grain sizes in Fig. 4b. Consequently, how did to distinguish these? It is also unclear, whether wm1 (muscovite porphyroclasts) could have been in dated aggregate. Show and discuss also the Ar release pattern of sample IO18-01, for which also Late Jurassic ages are listed in Table 1. (4) I recommend check the validity of of the apparent Late Cretaceous Ar-Ar ages by the isotope inversion. At least for samples AG3-03 and AG-3-5, there are potentially sufficient steps for such a task.

Some further issues: In Table 1, add mineralogy to each sample, this is a critical information. Show shear senses of various stages in the lower detailed map.

Specific and editorial remarks: L. 16: "southwards of the surface outcrop of the sub-duction megathrust: I find this an unlucky expression. Please reformulate. Not the surface outcrop was displaced, but the rock unit of the Asteroussia nappe. L. 20: For the informed reader, explain in the Introduction, why these rocks on Ios are (pre-Alpine) basement. This seems nowhere said with sufficient clarity. L. 36: For clarity, please, add for what this is a significant modification. Figure 2a, b: To accrete the Aegean terrane stack, this must have been accreted above the lower plate. L. 91-92: Refer to Fig. 3: "focusing on the north-west corner of the basement terranes in an attempt to determine the meaning of the previously reported 70-80 Ma ages." L. 105: "the Port Beach tectonic slice": not explicitly on Fig. 3 = Port Beach augengneiss? Fig. 3: Show location of lower graph on the Ios map above. Which fault is the the South Cyclades Shear Zone mentioned in the text? Ios fault? Correct in lower map legend "metemorphosed". L. 111-112: Show the field evidence of these multiple alternating deformation events mentioned here. Fig. 4a: Show the shear sense in figure: "(a) $\delta$-type garnet porphyroblasts in Port Beach garnet-mica schist." (c) What do you mean

with "not shown" in caption? "Large (1-2 mm muscovite fish ....not shown". Mention also which sample. Fig. 5: "two types small, second generation garnets with different chemical compositions are identified": The photomicrograph is convincing but show also the chemical composition of garnet core and rim. The second generation of garnet in (b) and (c) is greenish and dark and somehow untypical for usual garnet. Please provide more information. However, it is clear that the first and second generations are sharply separated and show two-stage growth. Mention also which sample. L. 128-129: "The prominent structural contact between the garnet-mica schist and the augengneiss is defined by a late-developed intense north-sense shear zone": Does this mean, that the augengneiss-garnet micaschist is another shear zone not shown on Fig. 3? L. 143-144: "suggesting P-T conditions of 100-140 MPa and 500-600oC": Specify the P-T conditions in a clear way how you reached these P-T conditions. L. 158: Mention why the garnet rim is black. L. 159-160: Are there compositional data on white mica inclusions within garnet? L. 171: Explain why not these are no "end member garnets". L. 175: "white mica–rutile inclusion": Composition of the white mica inclusions in garnet? L. 181-182: Use presence: "tectonic slice also records a complex history" L. 179-181: "The Si-content of white micas in sample IO17-03 and IO18-05 (both as mixture of muscovite and phengite) suggests that the phengite grew under P-T condition up to 500-1000 MPa and 400-500oC (Fig. 6c)": How did you calculate the P-T conditions? It remains also unclear at which metamorphic stage. Fig. 7: (a) There is a hornblende porphyroblast correctly labelled, but no white mica porphyroblast. "(a) an older generation white mica preserved as large porphyroclasts wrapped by younger, recrystallized white mica. In caption correct "whist".

L. 209-210: Refer to Supplement with Analytical details for 40Ar/39Ar dating. L. 223-224: "The argon geochronology analyses yielded age clusters in Early-Middle Jurassic, Late Cretaceous, Eocene–Oligocene and Oligocene–Miocene time (Table 1)": In fabrics, you distiguish between three white mica populations. L. 227-229: "The phengitic components produce significantly high activation energy estimates": Show and refer to the corresponding additional figure: This interpretation seems critical for the whole

story. Table 2: Explain N.A. L. 234-235: Aword seems missing in this sentence. Figure 7: Mention that these spectra are from a previous paper (Forster and Lister, 2009). L. 253: Correct "ourtcrops" L. 274-275: Reformulate: "potassium feldspar was replaced by metamorphic and/or metasomatic events at those times": Supposedly K-feldspar is still K-feldspar but recrystallized. L. 283: Better "K-feldspar concentrate" than "K-feldspar grain sample" L. 300-301: "The Ios data is the first report of Asteroussia ages in a terrane of unmistakeably Gondwanan affinity.": Add a reference for the Gondwanan affinity. L. 301-307: Most data are rather low-pressure, meaning T-dominated metamorphism.

Table 2: The headline "TABLE 2 – PUBLISHED PEAK METAMORPHIC CONDITION ESTIMATES IN THE CYCLADES" is clearly misleading. I suppose that yoe mean the P-T condition of the Asteroussia event. I recoomend also to ages if available. Then: "*with error on temperatures in the range of $\pm$ 50 °C": Add the error to the corresponding T estimates.

L. 326-328: Add references to this statement: "Dispute arises because of the focus on the exhumation of the Cycladic eclogite-blueschist terranes" L. 362: "Tripoliz": You mean Tripolitza? L. 364: Change slightly to: "This must have occurred sometime between "mid" Oligocene and early Miocene" There is no formal Middle Oligocene in the International Stratigraphic Chart. L. 365: Relation to eastern Alps seems unlikely: In eastern Alps, the late Cretaceous event is related to extension and exhumation of HP/UHP units.

References: Complete referencing: L. 488-489, L. 535, omit IF in L. 547

Supplementary material: Add the reference to the Flux monitor GA1550 (Spell & Mc-Dougall, 2003). I could not open the data tables. These must be included in the Supplementary Material.

---

## Referee Comment (RC2) · Anonymous Referee #2 · 15 Feb 2021

Dear Editor, dear Sonia Yeung and Co-authors,

First of all, I did not finalize the review until the discussion part, because I think the manuscript has to be rewritten fundamentally. My impression is, that the co-authors either did not check the manuscript in terms of literacy, organization and presentation or their comments were not acknowledged. The manuscript is poorly prepared, has language issues that make reading difficult, it is not concise and explicit, it has incomplete and illegible figures, figure captions, and the tables are improperly formatted. The structure of the manuscript is not clear. The separation between original and recycled data is not clear. Many statements about mineral chemistry remain unproven or not documented. Analytical errors are not shown in the mineral chemistry graphs. The tectonic background, introducing and differentiating between several deformation (D1,

D2, ect) events and the assignment of the different mineral generations (wm1, wm2, ect.) to those events is not done. A proper structural map or cross section is missing. The methodology is pooly described in terms of sample preparation, sample size, cleaning and measuring procedure (e.g., blank values and standard age values are not reported). In many paragraphs the headings are misleading and do not fit to the text. Therefore, I stopped before adding comments to the discussion section. In general, I do think the manuscript is suitable for a publication in SOLID EARTH and I do think after an intense rework, by the co-authors in the first place, it can make a substantial contribution to the scientific discussion.

All the best, Anonymous Reviewer 1

Please also note the supplement to this comment:
https://se.copernicus.org/preprints/se-2020-186/se-2020-186-RC2-supplement.pdf

[Figure]

**Supplement:**

[revised manuscript text omitted]

Mylopotas tectonic slice (structurally mid-level in Ios lower plate)

[revised manuscript text omitted]

---

## Author Comment (AC1) · 10 May 2021

**Evidence for the Late Cretaceous Asteroussia event in the Gondwanan Ios basement terranes**

Sonia Yeung[1], Marnie Forster[1], Emmanuel Skourtsos[2], Gordon Lister[1]

Authors' responses to the reviewer's comment.
Authors' responses are identified as blue text

**General comments**

(1) EMPA data is missing in the manuscript in addition to BSE images to distinct complex microfabrics dated.

The EMPA analysis and BSE images will be included in the supplementary materials of the resubmitted manuscript.

(2) Reviewer suggests considering analysis on garnets for P-T calculation.

We acknowledge that there is potential of extracting further information on fabric within the garnets for P-T calculation. But this cannot be done for this paper.

(3) P-T conditions for the Late Cretaceous Asteroussia event is generalized, detailed information on how this is derived is needed.

We will include pressure and temperature information that can be extracted from phengite silica content in the resubmitted manuscript.

(4) Distinction of the various white mica generations needs to be more detail, with information on how to identify them in the argon data.

We will revisit how this procedure is presented in the text and add details on the figure to show how different white mica generations can be recognised using the Arrhenius plot.

(5) Reviewer suggests checking the validity of the apparent Late Cretaceous Ar/Ar ages by the isotope inversion.

Analysis based on the York plot will be included in the resubmitted manuscript.

(6) Mineralogy of each sample should be added in table 1

Table 1 will be revised in the resubmitted manuscript.

(7) Shear senses of various stages of deformation needed to be shown in figure 3b.

Showing shear senses of various stages in the lower detailed map is difficult as the entire field area is affected by the broad, large scale shear zone. We will consider achieving this by including a field analysis with tectonic sequence diagrams (TSDs).
* * *
Specific and editorial remarks:

Reformulate the statement "southwards of the surface outcrop of the sub- duction megathrust" in line 16.

This sentence is reformulated in the revised manuscript.

The introduction does not explain clearly why rocks mentioned in line 20 are considered the (pre-Alpine) Ios basement.

We will revise this sentence with special note to this comment. This is a widely accepted understanding in previously published papers, but we will make it clearer in the resubmitted manuscript.

No explanation on why changes mentioned in line 36 is a significant modification.

We will reformulate this sentence with respect to this aspect.

Some detailed structural aspect is not clear in figure 2a, b, hence causes confusion.
The Aegean terrane stack has been accreted above the lower plate in figure 2a. We will enlarge the figure and potentially separate it into two so the details are clear in the resubmitted manuscript.

Sentence in line 91-92 needs to show reference to figure 3
This sentence is reformulated as suggested.

Tectonic unit "the Port Beach tectonic slice" mentioned in line 105 is not shown explicitly in figure 3.
The Port Beach tectonic slice include augengneiss (the main lithology) and other lithologies such as garnet-mica schist, graphitic quartzite and other smaller units. This 'tectonic slice' is used so as not to oversimplify the nature of this tectonic unit. We will reformulate the sentence to present the idea explicitly in the revised manuscript. A revised map will be provided.

Show location of the South Cyclades Shear Zone in figure 3 as mentioned in the text.
The South Cyclades Shear Zone overprints the entire field area with some places affected by the overprinting, narrow north-directed shear zone. This observation is added to the figure caption to ensure clear understanding.

"metemorphosed" is spell incorrectly in figure 3b map legend.
Map legend corrected in revised manuscript.

Field evidence of the multiple alternating deformation events mentioned in line 111-112 is missing.
We agree with the reviewer, a new figure will be added.

Sample number and labelling is unclear in figure 4a.
Corrections and modifications have been made to be clear and concise.

Figure 5 requires more information on the microstructures identified and analysed.
Corrections and modifications have been made which include further information, both in text and in the supplementary material in the revised manuscript.

The statement "The prominent structural contact between the garnet-mica schist and the augengneiss is defined by a late-developed intense north-sense shear zone" in line 128-129 is unclear.
We will reformulate this particular sentence.

Calculations and deduction to the P-T conditions reported in line 143-144 is unclear.
The resubmitted manuscript will include these details in the main text, with reference to literature documenting the method used in our calculation.

No explanation as to why the garnet rim is black in line 158.
The resubmitted manuscript will add further detail to this observation. Analytical data will be included in the supplementary material.

No compositional data on white mica inclusions within garnet is provided in line 159-160.
The grains are too small for this type of probe analysis. The garnet crystals are small in size (1-2 mm diameter), hence the inclusion will be even finer and compositional analysis unreasonable.

No explanation why "non-end member garnets" is used in line 171.
This particular sentence is restructured with additional information.

No composition data of the white mica inclusions in garnet in line 175.
Please see response to line 159-160

Presence tense is needed in line 181-182
This is corrected in the revised manuscript.

Method of the P-T conditions estimate is unclear and missing in line 179-181.
The resubmitted manuscript will include these details with reference to previous literature.

Correct a typo in figure 7(a).
This is corrected in the revised manuscript.

The reviewer suggests referring line 209-210 to the supplement with Analytical details for 40Ar/39Ar dating.
We agree with the reviewer.

Description of argon geochronology result in line 223- 224 needs to be similar to what described in the microstructure analysis.
We agree with the reviewer.

The reviewer suggests referring line 227-229 to the corresponding figure.
We agree with the reviewer.

No explanation on "N.A" in Table 2.
This is corrected in the revised manuscript.

A word is missing in line 234-235.
This is corrected in the revised manuscript.

The reviewer suggests mention that the spectra in figure 9 are from a previous paper (Forster and Lister, 2009).
This is corrected in the revised manuscript.

Correct "ourtcrops" in line 253.
This is corrected in the revised manuscript.

The reviewer suggests reformulating "potassium feldspar was replaced by metamorphic and/or metasomatic events at those times" in line 274-275.
We agree with the reviewer, the sentence is modified in the revised manuscript.

The reviewer suggests replacing "K-feldspar concentrate" in line 283 with "K- feldspar grain sample"
This is corrected in the revised manuscript.

A reference for the Gondwanan affinity is missing in line 300-301.
We agree with the reviewer and have revised this sentence.

The reviewer commented that: Most data are rather low-pressure, meaning T-dominated metamorphism' to lines 301-307.
Outcrops of the Asteroussia terrane in other Cycladic islands do overall demonstrate a high temperature – low pressure metamorphic condition. But the rocks in Ios suggests medium temperature – high pressure conditions. We will reformulate lines 301-307 to ensure clear presentation.

Title of table 2 is misleading.
The resubmitted manuscript will revise the table title with respect to this comment.

Reference is missing in line 326-328
This is corrected in the revised manuscript

Change "Tripoliz" to "Tripolitza" in line 362.
This is corrected in the revised manuscript

The reviewer suggests revising line 364 as there is no formal Middle Oligocene in the International Stratigraphic Chart.
This is corrected in the revised manuscript.

The reviewer thinks the relation to eastern Alps as presented in line 365 is unlikely.
We will fix this sentence. Note that this statement is listed as unresolved issues in this paper. We think it is worth noting such correlation when one is looking at the tectonic setting of this region (e.g., the extent of the Asteroussia terrane as reconstructed by Van Hinsbergen et al., 2020). The nature of the late Cretaceous event in the Eastern Alps is not discussed in that Greater Adria reconstruction paper. Yet, the idea has been presented and we therefore include it in the "unresolved issues" section.

Incomplete referencing in line 488-489, line 535
This is corrected in the revised manuscript.

Omit "IF" in line 547
This is corrected in the revised manuscript.

Reference to the Flux monitor GA1550 (Spell & Mc- Dougall, 2003) is missing in the supplementary material.
This is corrected in the revised manuscript.

The reviewer cannot open data tables in the Supplementary Material.
We will make sure these are included and accessible.

---

## Author Comment (AC2) · 10 May 2021

**Evidence for the Late Cretaceous Asteroussia event in the Gondwanan Ios basement terranes**

Sonia Yeung[1], Marnie Forster[1], Emmanuel Skourtsos[2], Gordon Lister[1]

Authors' responses to the reviewer's comment.
Authors' responses are identified as blue text

**General comments**

The manuscript is not concise and explicit
We will review the manuscript prior to resubmission with attention to its brevity and ensuring concise and explicit links to relevant material.

The manuscript has incomplete and/or illegible figures
We will review the manuscript prior to resubmission with attention to this aspect. We note that the submitted version had no problems in respect to incomplete or illegible figures.

The manuscript has incomplete figure captions
We will review the manuscript prior to resubmission with attention to this aspect.

Tables in the manuscript are improperly formatted
We will review the manuscript prior to resubmission with attention to this aspect.

The separation between original and recycled data is not clear.
We will review the manuscript prior to resubmission with attention to this aspect. Previously published data was explicitly identified in the submitted manuscript. However, it was not made clear that the submitted manuscript had new figures based on the analysis of these data.

Many statements about mineral chemistry remain unproven or not documented.
We will provide the data for mineral chemistry in the supplementary material.

Analytical errors are not shown in the mineral chemistry graphs.
We will review the manuscript prior to resubmission with attention to this aspect.

The manuscript does not provide tectonic background
This is not entirely correct – please see the section in the introduction. The resubmitted manuscript will more explicitly draw attention to the tectonic background as a separate section.

The manuscript does not differentiate deformation events (D1, D2, etc.)
This is correct, for the reason given in Forster and Lister (2008). Geochronology has shown that the tectonic evolution of the Cyclades is far more complex than the current simple $D_1$, $D_2$, $D_3$ scheme allows. We will mention this in the introduction. The paper has tried not to get entangled in the consequences of this failure, though we could suggest reform of current structural geology methodology is necessary.

The manuscript does not assign the different mineral generations (wm1, wm2, ect.) to D1, D2, etc.) events.
This is correct, as given above, but also because there is no reason that an episode of metamorphic mineral growth should *a priori* be linked to a deformation event. The paper uses Tectonic Sequence Diagrams (TSDs), which focus on reporting observational data. TSDs used in the paper include observed microstructural sequences linked to outcrop related sequences of deformation and metamorphic mineral growth events. Thereby we avoid model-based assumptions.  The resubmitted manuscript will draw more attention to this aspect.

A proper structural map or cross section is missing.
The structural map and cross-section are published in Forster et al. 2009. The resubmitted manuscript will explicitly draw attention to this point.

The methodology is poorly described in terms of sample preparation, sample size, cleaning and measuring procedure (e.g., blank values and standard age values are not reported).
We have reviewed the manuscript with attention to this aspect. To ensure the storyline of the paper, detailed description of the methodology will be placed in the supplementary material with the manuscript explicitly drawing attention to this supplementary material.

In many paragraphs the headings are misleading and do not fit to the text.
We have reviewed the manuscript with attention to this aspect.
* * *
**Specific and editorial remarks:**

Abstract:

The abstract is not explicit enough and in parts speculative.
All authors will be consulted, and the abstract will be revised in the resubmitted version.

Also the phrase that "Ar geochronology...demonstrates...metamorphic event" is misleading, metamorphic events are determined by petrology and theses events might be dated by geochronology.
The sentence will be revised in the resubmitted version.

Introduction:
The reviewer mentioned different aspects of the introduction that required further editing. We provide individual comment to each of the aspect and include our suggested structural change in the introduction to acknowledge the opinion of the reviewer.

The introduction does not introduce the main problem and ongoing discussion in literature adequately.
We agree with the reviewer and the introduction will be revised in the resubmitted version.

The introduction does not introduce the controversy in interpreting Ar-data of HP rocks.
We agree with the reviewer and the introduction with special attention to this aspect.

The introduction does not introduce what plates/ terranes are involved while some information are provided in figure 2 only.
Same comment as above, we will reformulate the text to give some connection between the figure and the text early in the paper.

The "Late Cretaceous metamorphic event" is not clear in the introduction (line 22).
The focus of this sentence is the identification of a metamorphic event at the time of Late Cretaceous Asteroussia age. Base on the reviewer's suggestion, we now changed it to "Late Cretaceous high pressure, medium temperature metamorphic event" to be more specific.

The revised Introduction will take specific note of the following in response to the reviewer's comment:
(1) The significant knowledge gap that exists in understanding old events in the Ios basement terrane prior to Alpine deformation events.
(2) The consequences of accreting this Gondwanan terrane to the Alpine terrane stack such as the extreme crustal extension after accretion, followed by magmatic event in Oligocene-Miocene period lack proper explanation.
(3) Until this work, it is largely assumed that the Ios basement is not affected by pre-Alpine deformation since ~300 Ma (hence referred as the Hercynian basement).
(4) Previous data did not recognize high-pressure rocks in the Ios basement and its tectonic history is much more complicated than a single $M_0$ event can define.
(5) Referring all older events as $M_0$ (pre-HP metamorphic event) results in little to no attention on older events prior to accretion in the evolution of the European terrane stacks
(6) Recognition of the exhumed Asteroussia terrane across the terrane stack in this study enabled us to identify a subduction jump that is impossible without a tectonic mode switch.
(7) Our proposed model on the subduction jump is able to capture and explain the extreme extension after accretion, formation of Cycladic metamorphic core complexes and the later Oligocene-Miocene magmatic event.

The Asteroussia Nappe (section 2):
The reviewer mentioned different aspects of this specific introduction section to the Asteroussia Nappe that requires further editing. We address each of the comment as follow. These aspects will be considered and revised in the resubmitted version.

The idea of "the same metamorphic age" outcropped in various Cycladic island is unspecified (line 58)
This particular sentence follows the deduction from line 54, discussing outcrops of the Asteroussia nappe across the Greek Cyclades. The main focus is the late Cretaceous 70-80 Ma age observed across all outcrops of the Asteroussia terrane (as summarised in table 2). The resubmitted manuscript will link this sentence to table 2 to make the idea more concise.

The published Rb-Sr dates mentioned does not include details to its interpretation (line 61).
We agree with the reviewer, the revised manuscript will pay special attention to the background behind such statement for better presentation.

The reviewer suggests replacing "low-pressure " by "retrograde" in line 65.
We found "overprinted by low-pressure greenschist facies" better to describing the occurrence of a younger deformation event with different property instead of saying that the rock experienced "retrograde" metamorphism due to the complexity of deformation and mineral growths associated. All authors will be consulted, and the sentence will be revised in the resubmitted version.

The reviewer suggests reformatting section 2.2 (which include the main research hypothesis/question) to link it with the Introduction and integrate research methodology in this section.
We agree with the reviewer, details to the proposed introduction are discussed above. The resubmitted manuscript will consider linking section 2.2 to detailed research methodology in the supplementary material.

The reviewer suggests adding references in line 77-78.
We agree with the reviewer, this has been corrected.

This section failed to present the field results demonstrating different deformation events
The revised manuscript will include details to explicitly draw attention to structural results on the various deformation events.

Tables of quantitative white mica data is missing in this section
The revised manuscript will provide connection between the text and the analytical results in the supplementary material.

**Microstructural analysis and mineral chemistry (section 3):**
The reviewer suggested various changes to this result section. We address each of the comment separately and will revise the raised concern in the resubmitted version.

A large portion of this paragraph should be shifted into the former section to describe and introduce the deformation zones and deformation phases.
We agree with the reviewer, the resubmitted manuscript will pay special attention to this recommendation. Section 2.2 (The Asteroussia event on Ios) will be revised:
  (1) Introduction of the four terrane slices in Ios. This will include a summary of the literature review and controversies on the structure of the terrane stack outcropped in Ios
  (2) Deformation zones and events recorded in this study will be presented
  (3) The section will conclude why the identification of the Asteroussia event on Ios will provide a significant knowledge advancement in the tectonic architecture in Ios and the Cyclades, hence this paper.

Quantitative data should be given in tables, e.g. in supplements
The revised manuscript will include text in this section explicitly drawing attention to the supplementary material with quantitative data and its methodology.

The reviewer suggests shifting the sentence in line 100-103 to the introduction.
We agree with the reviewer and have changed accordingly.

The reviewer suggests shifting the sentence in line 104-106 to section 2 – the Asteroussia nappe.
We agree with the reviewer and have changed accordingly.

The reviewer suggests enlarging the text in figure 3.
We agree with the reviewer and have changed accordingly.

The reviewer suggests shifting the sentence in line 111-112 to section 2 – the Asteroussia nappe. The reviewer also suggests providing field evidences to this sentence.
We agree with the reviewer regarding the new location of this sentence. If we show the field evidence of these event, this will entail the addition of a new figure, altering the entire manuscript to suit.

The reviewer suggests specifying the term "both tectonic silvers" in line 113-114.
We agree with the reviewer and have changed accordingly.

The reviewer suggests re-organising table 1 to be more space efficient.
We agree with the reviewer and have changed accordingly.

The reviewer suggests introducing all metamorphic events (including the "$\Delta_{1D}$ event" in Forster et al. (2020)) in the introduction.
We agree with the reviewer. Revision to the introduction is discussed above.

The reviewer suggests specifying the deformation event associated to the described deformation in line 124-125.
We agree with the reviewer and have changed accordingly.

The reviewer suggests providing structural description in terms of a map and cross section(s) to illustrate the deformation event associated to the described deformation in line 125-127.
We agree with the reviewer. We will adjust and correlate this sentence to in text figures and supplementary materials accordingly.

The two white mica generations need to be clearer in figure 4 with added sample details.
We acknowledge the reviewer's opinion. We will add details to this part (both in text and as figure – higher resolution image) to provide further details to the observation in the resubmitted manuscript.

Figure 4c need more statistical information.
We acknowledge the reviewer's opinion. The data presented in this chart is the exact value calculated from experimental data, but we will provid calculation details and background in text.

Differentiation of the different white mica generations is not concise in line 140-142.
We acknowledge the reviewer's opinion. We will include a detailed photo illustrating the crosscutting relation of the two white mica generations.

The concluding sentence in line 145-146 need evidence.
In the previous sentence, we provide an estimated peak metamorphic P-T condition based on phengite Si-content. This led to our "conclusion" in this sentence. We will be more specific in the sentence by provider a stronger deductive linkage in the resubmitted manuscript.

Sample number is needed in Figure 5 caption.
We agree with the reviewer. This is corrected in the revised manuscript.

Further specification is needed in describing the "non-end member" garnet in line 154-155.
We agree with the reviewer. We will provide further information in the supplementary material and directions to refer to such result in text.

Further specification is needed in differentiate between two iron ions in composition calculation in line 155-156, and data is to be supplied.
We agree with the reviewer. We will reformulate the statement and provide the data in the supplementary material.

Words that are 'interpretations' should be corrected to words that 'present' the result in line 156-157.
We agree with the reviewer, the sentence is corrected accordingly in the revised manuscript.

Replace "shear zone operation" by "shearing" in line 161-162.
We agree with the reviewer, this is corrected in the revised manuscript.

Rephrase line 162-163 to include argument for more intense deformation mentioned in text.
We attempt to compare between deformation fabrics in sample IO17-03 and IO18-05. The sentence is rephrased.

Data presented in figure 6c needs further specification on statistical details
The error bars are initially omitted in the figure to avoid confusion as some data points cluster tightly together. This will be discussed among all authors.

A change of line spacing in line 170
This is corrected in the revised manuscript.

The reviewer think it is unnecessary to highlight the idea of "non-endmember garnet" in line 171-172 as most garnets are mixed crystals.
We agree with the reviewer and now describe the garnets as "garnet porphyroblasts with chemical composition between almandine -grossular" and refer to data in the supplementary material.

The reviewer thinks the differentiation between these two groups is not convincing in line 180-181. And ask for reference to the barometer used.
We acknowledge the reviewer's suggestion. The sentence will be re-structured in the resubmitted manuscript. References of the used barometer are added, how the result is deduced will also be included in text, with reference to the supplementary material if needed.

The structural unit and sample name of the sample is missing in figure 7
We agree with the reviewer and have added the suggested details to the figure caption.

Line 190-192 should be shifter to section 2 to describe observation in a more systematic way.
We acknowledge the reviewer's suggestion and will revise the sentence in such the information is (1) mentioned in section 2 and (2) relates observation in section 2 to analytical results presented in this section (section 3).

The phrase "Thin-section parallel to the stretching lineation" in line 194 needs clarification and should be added to the figure caption of figure 7.
We acknowledge the reviewer's suggestion and will revise the sentence accordingly.

The reviewer suggests the connection of "quartz filled cracks created by crustal stretching" in line 196 is an interpretation that should go into the discussion.
We agree with the reviewer that this is indeed an interpretation hence need reformatting in the result section of the paper. This suggestion will be considered when reformulating the resubmitted manuscript accordingly.

The reviewer suggests adding details on sample preparation as artificial small grains might be resulted when reducing a rock sample to grains in line 220.
We acknowledge the reviewer's opinion. We will provide more details and reasoning in methodology selection in the supplementary material with some connection between the text and the supplementary material. The diffusion domain for argon can be significantly smaller than the grain radius, hence artificial small grains is unlikely to cause significant impact on data quality.

The reviewer suggests reformatting line 210-211 to be more explicit.
We agree with the reviewer and will reformulate this particular sentence in the resubmitted manuscript.

---

## Author Response (AR1)

**Evidence for the Late Cretaceous Asteroussia event in the Gondwanan Ios basement terranes**

Sonia Yeung[1], Marnie Forster[1], Emmanuel Skourtsos[2], Gordon Lister[1]

Authors' responses to the reviews including a list of all relevant changes made in the resubmitted manuscript. Authors' responses are identified as blue text

Overall comment

**The manuscript is not concise and explicit**

The manuscript is reviewed prior to resubmission with attention to its brevity and ensuring concise and explicit links to relevant material. Rearrangements of sentence and content are made to address this issue and facilitate logical flow of the manuscript.

**The manuscript has incomplete and/or illegible figures**

Figures are enlarged in the resubmitted manuscript to address this comment. We note that the resubmitted version had no problems in respect to incomplete or illegible figures.

**The manuscript has incomplete figure captions**

Figure captions in the resubmitted manuscript are reviewed and edited with attention to this aspect.

**Tables in the manuscript are improperly formatted**

Tables are reformatted in the resubmitted manuscript.

**The separation between original and recycled data is not clear.**

Previously published data was explicitly identified in the resubmitted manuscript. The resubmitted manuscript had new figures based on the analysis of these data and are identified both intext and in the supplementary material.

**The manuscript does not provide tectonic background**

The Introduction of the resubmitted manuscript is restructured to explicitly draw attention to the tectonic background.

**The manuscript does not differentiate deformation events (D1, D2, etc.)**

As this paper tried not to get entangled in the consequences of the numbering scheme failing to document the complex tectonic evolution of the Cyclades, the D1, D2, D3 classification is not used. Instead, a new figure (figure 4) is included to illustrate this and report field observation on deformation events in the resubmitted manuscript.

**The manuscript does not assign the different mineral generations (wm1, wm2, etc.) to D1, D2, etc.) events.**

The paper uses Tectonic Sequence Diagrams (TSDs) instead of the numbering method mentioned in the reviewer's comment for the reason above. The TSDs focus on reporting observational data as there is no reason that an episode of metamorphic mineral growth should a priori be linked to a deformation event. A new figure (figure 4) is included to illustrate this and report field observation on deformation events in the resubmitted manuscript.

**A proper structural map or cross section is missing.**

The structural map and cross-section are published in Forster et al. 2009. The resubmitted manuscript draw attention to this reference.

**In many paragraphs the headings are misleading and do not fit to the text.**
We have reviewed and reformatted the manuscript with attention to this aspect.

**Abstract**
**The abstract is not explicit enough and in parts speculative.**
All authors are consulted, the abstract was slightly modified in the resubmitted version in response to this comment.

**The phrase "Ar geochronology...demonstrates...metamorphic event" in the abstract is misleading, metamorphic events are determined by petrology and these events might be dated by geochronology.**
The sentence was revised in the resubmitted manuscript.

**Section 1**
**The introduction does not introduce the main problem and ongoing discussion in literature adequately.**
The introduction was restructured with the following details in the resubmitted manuscript:
  (1) The significant knowledge gap that exists in understanding old events in the Ios basement terrane prior to Alpine deformation events.
  (2) The consequences of accreting this Gondwanan terrane to the Alpine terrane stack such as the extreme crustal extension after accretion, followed by magmatic event in Oligocene-Miocene period lack proper explanation.
  (3) Until this work, it is largely assumed that the Ios basement is not affected by pre-Alpine deformation since ~300 Ma (hence referred as the Hercynian basement).
  (4) Previous data did not recognize high-pressure rocks in the Ios basement and its tectonic history is much more complicated than a single $M_0$ event can define.
  (5) Referring all older events as $M_0$ (pre-HP metamorphic event) results in little to no attention on older events prior to accretion in the evolution of the European terrane stacks
  (6) Recognition of the exhumed Asteroussia terrane across the terrane stack in this study enabled us to identify a subduction jump that is impossible without a tectonic mode switch.
  (7) Our proposed model on the subduction jump is able to capture and explain the extreme extension after accretion, formation of Cycladic metamorphic core complexes and the later Oligocene-Miocene magmatic event

**The introduction does not introduce the controversy in interpreting Ar-data of HP rocks.**
The introduction was revised with special concern in this aspect in the resubmitted manuscript.

**The introduction does not introduce what plates/ terranes are involved while some information are provided in figure 2 only.**
The text and figure 2 (including figure caption) are reformulated in the resubmitted manuscript in response to this comment.

**Reformulate the statement "southwards of the surface outcrop of the sub- duction megathrust" in line 16.**
This sentence is reformulated in the revised manuscript.

**The introduction does not explain clearly why rocks mentioned in line 20 are considered the (pre-Alpine) Ios basement.**
The introduction is restructured in the resubmitted manuscript, with special note to this comment.

**The "Late Cretaceous metamorphic event" is not clear in the introduction (line 22).**
The sentence is restructured in the resubmitted manuscript, with added details to explain the idea clearly.

**No explanation on why changes mentioned in line 36 is a significant modification.**
The sentence was reformulated and is part of the restructured introduction in the resubmitted manuscript.

**Section 2**
**The idea of "the same metamorphic age" outcropped in various Cycladic islands is unspecified (line 58)**
This particular sentence was revised in the resubmitted manuscript with elaborations on the idea of "the same (Late Cretaceous) metamorphic age" reported across the Cycladic islands.

**The published Rb-Sr dates mentioned does not include details to its interpretation (line 61).**
The revised sentence added details to the Rb-Sr dates such as the lithology analysed and its location in the Cyclades.

**The reviewer suggests replacing "low-pressure" by "retrograde" in line 65.**
We found "overprinted by low-pressure greenschist facies" better to describing the occurrence of a younger deformation event with different property instead of saying that the rock experienced "retrograde" metamorphism due to the complexity of deformation and mineral growths associated. Hence this sentence is not revised as suggested in the resubmitted version.

**The reviewer suggests reformatting section 2.2 (which include the main research hypothesis/question) to link it with the Introduction and integrate research methodology in this section.**
The resubmitted manuscript provides a better linkage between the introduction, section 2.2 and section 3. Details to the research methodology is included the supplementary material with linkage provided in the result section (section 3 and 4).

**The reviewer suggests adding references in line 77-78.**
References has been added to this particular sentence.

**Section 2.2 failed to present the field results demonstrating different deformation events**
The revised manuscript include details to explicitly draw attention to structural results on the various deformation events, this includes the structural maps provided in the supplementary material and the new figure 4.

**EPMA data is missing in the manuscript in addition to BSE images to distinct complex microfabrics dated.**
The EPMA analysis and BSE images are included in the supplementary materials of the resubmitted manuscript.

**Tables of quantitative white mica data is missing in this section**
See response above.

**Many statements about mineral chemistry remain unproven or not documented.**
Results from EPMA analysis are provided in the supplementary material, with in text notification pointing to the supplementary material.

**Analytical errors are not shown in the mineral chemistry graphs.**

Analytical errors are not placed in the mineral chemistry graphs for clearer identification of data points in the cluster. Exact values measured are presented in the supplementary material.

**Reviewer suggests considering analysis on garnets for P-T calculation.**

We acknowledge that there is potential of extracting further information on fabric within the garnets for P-T calculation. But this cannot be done for this paper.

**P-T conditions for the Late Cretaceous Asteroussia event is generalized, detailed information on how this is derived is needed.**

We included the literature used and logic used to derive our presented pressure and temperature extracted from phengite silica content in the resubmitted manuscript.

**Distinction of the various white mica generations needs to be more detail, with information on how to identify them in the argon data.**

We revisited how this procedure is presented in the text and added details on the figures to show how different white mica generations can be recognised. Additional information on recognition of different white mica generations using the Arrhenius plot and the York plot are presented in the supplementary material.

**The methodology is poorly described in terms of sample preparation, sample size, cleaning and measuring procedure (e.g., blank values and standard age values are not reported).**

We have reviewed the manuscript with attention to this aspect. To ensure the storyline of the paper, detailed description of the methodology is included in the supplementary material with the manuscript explicitly drawing attention to this supplementary material.

**Some detailed structural aspect is not clear in figure 2a, b, hence causes confusion.**

An enlarged figure is added to figure 2 for clear illustration of details in the resubmitted manuscript.

**Sentence in line 91-92 needs to show reference to figure 3**

This sentence is reformulated as suggested.

**Tectonic unit "the Port Beach tectonic slice" mentioned in line 105 is not shown explicitly in figure 3.**

The sentence and details in figure are revised in the revised manuscript.

**Show location of the South Cyclades Shear Zone in figure 3 as mentioned in the text.**

The South Cyclades Shear Zone overprints the entire field area with some places affected by the overprinting, narrow north-directed shear zone. This observation is added to the figure caption to ensure clear understanding.

**"metemorphosed" is spell incorrectly in figure 3b map legend.**

Map legend corrected in the revised manuscript.

**Shear senses of various stages of deformation needed to be shown in figure 3b.**

Showing shear senses of various stages in the lower detailed map is difficult as the entire field area is affected by the broad, large scale shear zone. We achieved this by including a new figure (figure 4 in the resubmitted manuscript) summarizing field analysis with tectonic sequence diagrams (TSDs) and compared with the traditional structural geology numbering method.

**Section 3**

**Field evidence of the multiple alternating deformation events mentioned in line 111-112 is missing.**

A new figure (figure 4) is added in the revised manuscript to provide field evidence requested. Structural maps revised from Yeung, 2019 Master's thesis are included in the supplementary material.

**Sample number and labelling is unclear in figure 4a.**

Corrections and modifications have been made to be clear and concise.

**Figure 5 requires more information on the microstructures identified and analysed.**

Corrections and modifications have been made which include further information, both in text and in the supplementary material in the revised manuscript.

**The statement "The prominent structural contact between the garnet-mica schist and the augengneiss is defined by a late-developed intense north-sense shear zone" in line 128-129 is unclear.**

This particular sentence is reformulated in the revised manuscript.

**Calculations and deduction to the P-T conditions reported in line 143-144 is unclear.**

The resubmitted manuscript included the literature used and logic used to derive our presented pressure and temperature extracted from phengite silica content in the resubmitted manuscript.

**No explanation as to why the garnet rim is black in line 158.**

The resubmitted manuscript added further detail to this observation. We do not know the cause of the black garnet rim and present in text that this might be due to a chemical composition change during the second mineral growth event. Analytical data will be included in the supplementary material.

**No compositional data on white mica inclusions within garnet is provided in line 159-160.**

The grains are too small for this type of probe analysis. The garnet crystals are small in size (1-2 mm diameter), hence the inclusion will be even finer and compositional analysis unreasonable.

**No explanation why "non-end member garnets" is used in line 171.**

This particular sentence is restructured with additional information.

**No composition data of the white mica inclusions in garnet in line 175.**

Please see response to reviewer's comment on line 159-160

**Presence tense is needed in line 181-182**

This is corrected in the revised manuscript.

**Method of the P-T conditions estimate is unclear and missing in line 179-181.**

The resubmitted manuscript includes these details with reference to previous literature.

**A large portion of this paragraph should be shifted into the former section to describe and introduce the deformation zones and deformation phases.**

Section 2.2 (The Asteroussia event on Ios) is reformatted in the resubmitted manuscript to create a better linkage to the introduction with the following structure:

(1) Introduction of the four terrane slices in Ios. This will include a summary of the literature review and controversies on the structure of the terrane stack outcropped in Ios

(2) Deformation zones and events recorded in this study will be presented

(3) The section will conclude why the identification of the Asteroussia event on Ios will provide a significant knowledge advancement in the tectonic architecture in Ios and the Cyclades, hence this paper.

**Quantitative data should be given in tables, e.g. in supplements**
The revised manuscript include text in this section explicitly drawing attention to the supplementary material with quantitative data and its methodology.

**The reviewer suggests shifting the sentence in line 100-103 to the introduction.**
This is changed accordingly in the resubmitted manuscript.

**The reviewer suggests shifting the sentence in line 104-106 to section 2 – the Asteroussia nappe.**
This is changed accordingly in the resubmitted manuscript.

**The reviewer suggests enlarging the text in figure 3.**
The figure is enlarged in the resubmitted manuscript.

**The reviewer suggests shifting the sentence in line 111-112 to section 2 – the Asteroussia nappe. The reviewer also suggests providing field evidences to this sentence.**
This change is made with the addition of a new figure (figure 4) in the resubmitted manuscript.

**The reviewer suggests specifying the term "both tectonic silvers" in line 113-114.**
Details that specify this term are added in the resubmitted manuscript.

**Mineralogy of each sample should be added in table 1**
Table 1 is revised accordingly in the resubmitted manuscript.

**The reviewer suggests re-organising table 1 to be more space efficient.**
Table 1 is changed accordingly in the resubmitted manuscript.

**The reviewer suggests introducing all metamorphic events (including the "$\Delta_{1D}$ event" in Forster et al. (2020)) in the introduction.**
The comment is accepted in the revised manuscript in the new figure (figure 4 in resubmitted manuscript).

**The reviewer suggests specifying the deformation event associated to the described deformation in line 124-125.**
This is changed accordingly in the resubmitted manuscript.

**The reviewer suggests providing structural description in terms of a map and cross section(s) to illustrate the deformation event associated to the described deformation in line 125-127.**
This sentence is adjusted and correlated to supplementary materials accordingly.

**The two white mica generations need to be clearer in figure 4 with added sample details.**
Further details to the observation in figure 4 (now figure 5) is provided in the resubmitted manuscript.

**Figure 4c need more statistical information.**
Data presented in this chart is the exact value calculated from experimental data, analytical data (EPMA data) used to calculate the figure 4c is included in the resubmitted supplementary material.

**Differentiation of the different white mica generations is not concise in line 140-142**.
Observations in this particular sentence can be referred to figure 5b and the figure caption in the resubmitted manuscript.

**The concluding sentence in line 145-146 need evidence.**
The sentence is restructured in the resubmitted manuscript with added details to the P-T estimation such as literatures used to calculate the estimated values.

**Sample number is needed in Figure 5 caption.**
This is corrected in the revised manuscript.

**Further specification is needed in describing the "non-end member" garnet in line 154-155.**
This sentence is restructured in the revised manuscript, further information is provided in the supplementary material.

**Further specification is needed in differentiate between two iron ions in composition calculation in line 155-156, and data is to be supplied.**
The sentence is reformatted to avoid confusion in the resubmitted manuscript.

**Words that are 'interpretations' should be corrected to words that 'present' the result in line 156-157.**
The sentence is corrected accordingly in the revised manuscript.

**Replace "shear zone operation" by "shearing" in line 161-162.**
This is corrected in the revised manuscript.

**Rephrase line 162-163 to include argument for more intense deformation mentioned in text.**
The sentence is rephrased.

**Data presented in figure 6c needs further specification on statistical details**
The error bars are initially omitted in the figure to avoid confusion as some data points cluster tightly together. This figure remains as it is in the resubmitted manuscript but raw data collected from the EPMA analysis is included in the supplementary material.

**A change of line spacing in line 170**
This is corrected in the revised manuscript.

**The reviewer think it is unnecessary to highlight the idea of "non-endmember garnet" in line 171-172 as most garnets are mixed crystals.**
This is corrected in the revised manuscript.

**The reviewer thinks the differentiation between these two groups is not convincing in line 180-181. And ask for reference to the barometer used.**
We acknowledge the reviewer's suggestion. The sentence is re-structured in the resubmitted manuscript. References of the used barometer are added so the equations used for calculating the result can be traced back to the literature.

**The structural unit and sample name of the sample is missing in figure 7**
The suggested details are added to the figure caption.

**Line 190-192 should be shifted to section 2 to describe observation in a more systematic way.**
The sentence is restructured in the revised manuscript.

**The phrase "Thin-section parallel to the stretching lineation" in line 194 needs clarification and should be added to the figure caption of figure 7.**

The sentence is revised in the revised manuscript.

**The reviewer suggests the connection of "quartz filled cracks created by crustal stretching" in line 196 is an interpretation that should go into the discussion.**
This suggestion is considered when reformulating the resubmitted manuscript accordingly.

Section 4
**Reviewer suggests checking the validity of the apparent Late Cretaceous Ar/Ar ages by the isotope inversion.**
This has been done and the process and logic involve is presented along with the York plots in the supplementary material of the resubmitted manuscript.

**Correct a typo in figure 7(a).**
This is corrected in the revised manuscript.

**The reviewer suggests referring line 209-210 to the supplement with Analytical details for 40Ar/39Ar dating.**
Such linkage has been made in the resubmitted manuscript.

**The reviewer suggests reformatting line 210-211 to be more explicit.**
The sentence is revised in the resubmitted manuscript.

**The reviewer suggests adding details on sample preparation as artificial small grains might be resulted when reducing a rock sample to grains in line 220.**
The sentence is revised in the resubmitted manuscript. More details and reasoning in methodology selection is provided in the supplementary material.

**Description of argon geochronology result in line 223- 224 needs to be similar to what described in the microstructure analysis.**
Argon geochronology result producing different age clusters are linked to various observed microstructures in the resubmitted manuscript.

**The reviewer suggests referring line 227-229 to the corresponding figure.**
The corresponding figure to the data presented in line 227-229 is presented in the supplementary material. Connection is made between the text and the supplementary material in the resubmitted manuscript.

**No explanation on "N.A" in Table 2.**
This is corrected in the revised manuscript.

**A word is missing in line 234-235.**
This is corrected in the revised manuscript.

**The reviewer suggests mention that the spectra in figure 9 are from a previous paper (Forster and Lister, 2009).**
This is corrected in the revised manuscript.

**Correct "ourtcrops" in line 253.**
This is corrected in the revised manuscript.

The reviewer suggests reformulating "potassium feldspar was replaced by metamorphic and/or metasomatic events at those times" in line 274-275.

The sentence is modified in the revised manuscript.

**Section 5**

The reviewer suggests replacing "K-feldspar concentrate" in line 283 with "K- feldspar grain sample"

This is corrected in the revised manuscript.

A reference for the Gondwanan affinity is missing in line 300-301.

The sentence is revised in the revised manuscript.

The reviewer commented that: Most data are rather low-pressure, meaning T-dominated metamorphism' to lines 301-307.

Additional information is added to ensure clear presentation in this part of the resubmitted manuscript.

Title of table 2 is misleading.

The table title is revised with respect to this comment.

Reference is missing in line 326-328

This is corrected in the revised manuscript

Change "Tripoliz" to "Tripolitza" in line 362.

This is corrected in the revised manuscript

The reviewer suggests revising line 364 as there is no formal Middle Oligocene in the International Stratigraphic Chart.

This is corrected in the revised manuscript.

The reviewer thinks the relation to eastern Alps as presented in line 365 is unlikely.

As this statement is listed as unresolved issues in this paper. We think it is worth noting such correlation when one is looking at the tectonic setting of this region (e.g., the extent of the Asteroussia terrane as reconstructed by Van Hinsbergen et al., 2020). The nature of the late Cretaceous event in the Eastern Alps is not discussed in that Greater Adria reconstruction paper. Yet, the idea has been presented and we therefore include it in the "unresolved issues" section in the resubmitted manuscript.

**References and supplementary material**

Incomplete referencing in line 488-489, line 535

This is corrected in the revised manuscript.

Omit "IF" in line 547

This is corrected in the revised manuscript.

Reference to the Flux monitor GA1550 (Spell & Mc-Dougall, 2003) is missing in the supplementary material.

This is corrected in the revised manuscript.

The reviewer cannot open data tables in the Supplementary Material.

The data tables are included and accessible in the supplementary material.

[revised manuscript text omitted]

| Van Der Maar, 1979 | Vandenberg and Lister, 1994 | Forster and Lister, 2009 | Correlation of TSDs developed in this study |
|---|---|---|---|
| Pre-Hercynian granite bodies intrusion | S₀ layering | | granitoid intrusion |
| M₁ Pre-Alpine Hercynian amphibolite facies ~295-305 Ma | D₁ deformation event: relicts of deformation preserved as S₁ microliths in S₂ | TSD's for earlier events not recorded | S₁/S₀ Δgrt (large porphyroblast) / S₂/S₁ Δbt wrapping grt / Sdiff (Scm) / Vein ctz (//S₁) Alteration / Δbt Δwm Δpy Δhem |
| M₂ Eocene HP–MT eclogite-blueschist facies ~ 42-59 Ma | D₂ folding event (crustal shortening) formation of F₂ folds with: S₂ axial plane cleavages N-S oriented L₂ lineation | Δ₄₉ growth event in eclogite-blueschist unit (EBU), 52-53 Ma, omphacite + jadeite | porphyroblast growth event Δgrt (large porphyroblast) |
| | | post Δ₄₉ SZ operation, 49-53 Ma | |
| | | F₃ (recumbent folding) | Fᴿ Spx  Fᴜ (folding) |
| | | Δ₄₃ porphyroblastic event in EBU, 43-45 Ma, epidote+glaucophane+garnet+mica | porphyroblasted growth event Δgrt (invading foam textures in quartz aggregates) |
| | | post Δ₄₃ SZ operation, 40-44 Ma | Dsz  Δwm (recrystallised fabric) |
| M₃ Oligo–Miocene greenschist facies ~ 25-16 Ma | D₃ recumbent folding event (crustal shortening) formation of F₃ folds which: folded S₂ and L₂, develops S₃ differentiation crenulation in axial zone | F₃ | Fᴜ Spx |
| | | Δ₃₅ porphyroblastic event in EBU, 34-35 Ma, transitional blueschist-greenschist facies biotite+garnet+mica | Δgrt light core of smaller, zoned porphyroblasts Δgrt dark rim of smaller, zoned porphyroblasts |
| | D₄ SCSZ operation (crustal extension) forms N-S oriented s₄ lineations | post Δ₃₅ SCSZ operation, 35-30 Ma | DᴺCSZ (south-directed) R (rotation of porphyroclasts) Δwm (recrystallised fabric) (S C plane fabrics) Δqtz (mantle) |
| M₄ Late Miocene Granitoid intrusion (LP-HT contact metamorphism) ~ 22-10 Ma | | | Fᴿeclined Fꟙ  Fᴜ (defined by qtz vein) (fluid limbs recognised as boudinage) Spx |
| | | intense North sense SZ operation, 25-29 Ma, rapid extension ~ 25 Ma with Δ₂₅ and Δ₂₄ mineral growth event under greenschist facies | DᴺCSZ (north directed) Δqtz (porphyroblast) Δwm (recrystallisation) R Δ
[revised manuscript text omitted]

Deleted: (INTERPRETATION?) It is unknown why the garnets are black, with a greenish hue. But with the Ios basement terrane affected by a pervasive green schist facies, it is possible that these black garnets are subjected to mineral replacement much like the chloritoid altered garnets in the Port Beach tectonic slice. Since focus of this research is on pre-Alpine deformation events, no correlation between garnets in these two outcrops are made as yet.

[Figure]

Figure 7: Microstructure analysis in the garnet-mica schist tectonic slice (sample IO17-03). (a) White mica dominated deformation fabric with minor biotite relict of early fabric surrounds large 2–3 cm diameter garnets, with younger 2–3 mm garnets grown on the deformation fabric. (b) Thin section under plane polarized light: two types small, second generation garnets with different chemical compositions are identified (see supplementary material – EPMA analysis results). (c). A slightly larger (~ 4mm diameter) second generation garnet with a zoned crystalline texture. (d) magnified view of the green box in (b), a euhedral second-generation garnet that grew into a quartz foam texture with relict rutile floating in the void space.

[Figure]

Si-content plot of white micas in the north Mylopotas garnet-mica schist

- ● Deformation fabric (groundmass) in IO17-03
- ◆ C' plane deformation fabric in IO18-05
- ◆ S plane deformation fabric in IO18-05

(a)

(b)

(c)

[revised manuscript text omitted]

Not Highlight

| Page 2: [6] Formatted | Margaret Forster | 10/06/2021 16:28:00 |
|---|---|---|

Not Highlight

| Page 2: [7] Formatted | Margaret Forster | 10/06/2021 16:28:00 |
|---|---|---|

Not Highlight

| Page 2: [8] Deleted | Gordon Lister | 25/06/2021 11:37:00 |
|---|---|---|

| Page 3: [9] Formatted | Gordon Lister | 25/06/2021 11:48:00 |
|---|---|---|

Not Highlight

| Page 3: [9] Formatted | Gordon Lister | 25/06/2021 11:48:00 |
|---|---|---|

Not Highlight

| Page 3: [10] Deleted | Gordon Lister | 25/06/2021 11:49:00 |
|---|---|---|

| Page 3: [10] Deleted | Gordon Lister | 25/06/2021 11:49:00 |
|---|---|---|

... [1]

... [2]

... [3]

Fo

Fo

Fo

Fo

**Page 3: [11] Deleted**      **Gordon Lister**      **25/06/2021 11:53:00**

**Page 3: [12] Deleted**      **Margaret Forster**      **10/06/2021 16:30:00**

**Page 3: [12] Deleted**      **Margaret Forster**      **10/06/2021 16:30:00**

**Page 3: [13] Deleted**      **Gordon Lister**      **25/06/2021 11:54:00**

**Page 3: [13] Deleted**      **Gordon Lister**      **25/06/2021 11:54:00**

**Page 3: [13] Deleted**      **Gordon Lister**      **25/06/2021 11:54:00**

**Page 3: [13] Deleted**      **Gordon Lister**      **25/06/2021 11:54:00**

**Page 3: [13] Deleted**      **Gordon Lister**      **25/06/2021 11:54:00**

**Page 3: [13] Deleted**      **Gordon Lister**      **25/06/2021 11:54:00**

**Page 3: [14] Deleted**      **Margaret Forster**      **10/06/2021 16:31:00**

**Page 3: [14] Deleted**      **Margaret Forster**      **10/06/2021 16:31:00**

| Page 3: [16] Deleted | Gordon Lister | 25/06/2021 12:00:00 |
| --- | --- | --- |

| Page 3: [16] Deleted | Gordon Lister | 25/06/2021 12:00:00 |
| --- | --- | --- |

| Page 3: [16] Deleted | Gordon Lister | 25/06/2021 12:00:00 |
| --- | --- | --- |

| Page 3: [16] Deleted | Gordon Lister | 25/06/2021 12:00:00 |
| --- | --- | --- |

| Page 4: [17] Deleted | Margaret Forster | 17/06/2021 13:49:00 |
| --- | --- | --- |

| Page 4: [17] Deleted | Margaret Forster | 17/06/2021 13:49:00 |
| --- | --- | --- |

| Page 4: [18] Deleted | Margaret Forster | 17/06/2021 13:50:00 |
| --- | --- | --- |

| Page 4: [18] Deleted | Margaret Forster | 17/06/2021 13:50:00 |
| --- | --- | --- |

| Page 4: [18] Deleted | Margaret Forster | 17/06/2021 13:50:00 |
| --- | --- | --- |

| Page 4: [19] Deleted | Gordon Lister | 25/06/2021 12:05:00 |
| --- | --- | --- |

**Page 4: [21] Deleted**     **Margaret Forster**      **17/06/2021 13:58:00**

**Page 4: [21] Deleted**     **Margaret Forster**      **17/06/2021 13:58:00**

**Page 4: [22] Deleted**     **Margaret Forster**      **17/06/2021 14:02:00**

**Page 4: [22] Deleted**     **Margaret Forster**      **17/06/2021 14:02:00**

**Page 4: [23] Deleted**     **Gordon Lister**      **25/06/2021 12:11:00**

**Page 4: [23] Deleted**     **Gordon Lister**      **25/06/2021 12:11:00**

**Page 4: [24] Deleted**     **Margaret Forster**      **17/06/2021 14:04:00**

**Page 4: [24] Deleted**     **Margaret Forster**      **17/06/2021 14:04:00**

**Page 4: [24] Deleted**     **Margaret Forster**      **17/06/2021 14:04:00**

Page 4: [26] Deleted          Margaret Forster          17/06/2021 14:43:00

Page 4: [26] Deleted          Margaret Forster          17/06/2021 14:43:00

Page 4: [26] Deleted          Margaret Forster          17/06/2021 14:43:00

Page 4: [27] Deleted          Gordon Lister          25/06/2021 14:07:00

Page 4: [27] Deleted          Gordon Lister          25/06/2021 14:07:00

Page 4: [27] Deleted          Gordon Lister          25/06/2021 14:07:00

Page 4: [27] Deleted          Gordon Lister          25/06/2021 14:07:00

Page 4: [27] Deleted          Gordon Lister          25/06/2021 14:07:00

Page 8: [28] Deleted          Ho Sonia Yeung          17/06/2021 18:24:00

Page 8: [29] Deleted          Gordon Lister          25/06/2021 14:20:00

**Page 8: [33] Deleted**      **Gordon Lister**      **29/06/2021 14:20:00**

**Page 8: [33] Deleted**      **Gordon Lister**      **29/06/2021 14:20:00**

**Page 8: [33] Deleted**      **Gordon Lister**      **29/06/2021 14:20:00**

**Page 8: [33] Deleted**      **Gordon Lister**      **29/06/2021 14:20:00**

**Page 8: [34] Deleted**      **Gordon Lister**      **29/06/2021 14:21:00**

**Page 8: [34] Deleted**      **Gordon Lister**      **29/06/2021 14:21:00**

**Page 8: [34] Deleted**      **Gordon Lister**      **29/06/2021 14:21:00**

**Page 8: [34] Deleted**      **Gordon Lister**      **29/06/2021 14:21:00**

**Page 8: [34] Deleted**      **Gordon Lister**      **29/06/2021 14:21:00**

**Page 8: [35] Deleted**      **Ho Sonia Yeung**      **17/06/2021 18:20:00**

**Page 8: [35] Deleted**      **Ho Sonia Yeung**      **17/06/2021 18:20:00**

**Page 8: [36] Deleted**      **Gordon Lister**      **29/06/2021 14:20:00**

**Page 8: [36] Deleted**      **Gordon Lister**      **29/06/2021 14:20:00**

**Page 8: [36] Deleted**      **Gordon Lister**      **29/06/2021 14:20:00**

**Page 8: [36] Deleted**      **Gordon Lister**      **29/06/2021 14:20:00**

**Page 8: [37] Deleted**     **Gordon Lister**     **29/06/2021 14:54:00**

**Page 8: [37] Deleted**     **Gordon Lister**     **29/06/2021 14:54:00**

**Page 8: [37] Deleted**     **Gordon Lister**     **29/06/2021 14:54:00**

**Page 8: [37] Deleted**     **Gordon Lister**     **29/06/2021 14:54:00**

**Page 8: [37] Deleted**     **Gordon Lister**     **29/06/2021 14:54:00**

**Page 8: [37] Deleted**     **Gordon Lister**     **29/06/2021 14:54:00**

**Page 8: [37] Deleted**     **Gordon Lister**     **29/06/2021 14:54:00**

**Page 8: [37] Deleted**     **Gordon Lister**     **29/06/2021 14:54:00**

**Page 8: [37] Deleted**     **Gordon Lister**     **29/06/2021 14:54:00**

**Page 8: [37] Deleted**     **Gordon Lister**     **29/06/2021 14:54:00**

**Page 8: [37] Deleted**     **Gordon Lister**     **29/06/2021 14:54:00**

**Page 8: [37] Deleted**     **Gordon Lister**     **29/06/2021 14:54:00**

**Page 11: [38] Deleted**     **Gordon Lister**     **30/06/2021 16:20:00**

**Page 11: [38] Deleted**     **Gordon Lister**     **30/06/2021 16:20:00**

Page 11: [38] Deleted          Gordon Lister          30/06/2021 16:20:00

Page 11: [38] Deleted          Gordon Lister          30/06/2021 16:20:00

Page 11: [38] Deleted          Gordon Lister          30/06/2021 16:20:00

Page 11: [38] Deleted          Gordon Lister          30/06/2021 16:20:00

Page 11: [38] Deleted          Gordon Lister          30/06/2021 16:20:00

Page 11: [38] Deleted          Gordon Lister          30/06/2021 16:20:00

Page 11: [38] Deleted          Gordon Lister          30/06/2021 16:20:00

Page 11: [38] Deleted          Gordon Lister          30/06/2021 16:20:00

Page 11: [38] Deleted          Gordon Lister          30/06/2021 16:20:00

Page 11: [38] Deleted          Gordon Lister          30/06/2021 16:20:00

Page 11: [38] Deleted          Gordon Lister          30/06/2021 16:20:00

**Page 11: [39] Deleted**   **Gordon Lister**   **30/06/2021 16:24:00**

**Page 11: [39] Deleted**   **Gordon Lister**   **30/06/2021 16:24:00**

**Page 11: [39] Deleted**   **Gordon Lister**   **30/06/2021 16:24:00**

**Page 11: [39] Deleted**   **Gordon Lister**   **30/06/2021 16:24:00**

**Page 11: [39] Deleted**   **Gordon Lister**   **30/06/2021 16:24:00**

**Page 11: [39] Deleted**   **Gordon Lister**   **30/06/2021 16:24:00**

**Page 11: [39] Deleted**   **Gordon Lister**   **30/06/2021 16:24:00**

**Page 11: [39] Deleted**   **Gordon Lister**   **30/06/2021 16:24:00**

**Page 11: [39] Deleted**   **Gordon Lister**   **30/06/2021 16:24:00**

**Page 11: [39] Deleted**   **Gordon Lister**   **30/06/2021 16:24:00**

**Page 11: [39] Deleted**   **Gordon Lister**   **30/06/2021 16:24:00**

Page 11: [40] Deleted        Gordon Lister        30/06/2021 16:35:00

Page 11: [40] Deleted        Gordon Lister        30/06/2021 16:35:00

Page 11: [40] Deleted        Gordon Lister        30/06/2021 16:35:00

Page 11: [40] Deleted        Gordon Lister        30/06/2021 16:35:00

Page 11: [40] Deleted        Gordon Lister        30/06/2021 16:35:00

Page 11: [40] Deleted        Gordon Lister        30/06/2021 16:35:00

Page 11: [40] Deleted        Gordon Lister        30/06/2021 16:35:00

Page 17: [41] Deleted        Gordon Lister        25/06/2021 14:25:00

Page 19: [42] Deleted        Ho Sonia Yeung       09/06/2021 22:55:00

Page 19: [43] Deleted        Ho Sonia Yeung       10/06/2021 08:52:00

Page 19: [44] Deleted        Gordon Lister        25/06/2021 14:26:00

Page 21: [45] Deleted        Gordon Lister        30/06/2021 17:35:00